



# Investigating oxygen and carbon isotopic relationships in speleothem records over the last millennium using multiple isotope-enabled climate models

Janica C. Bühler[1]★, Josefine M. Axelsson[2]★, Franziska A. Lechleitner[3], Jens Fohlmeister[4,5], Allegra N. LeGrande[6], Madhavan Midhun[7,8], Jesper Sjolte[9], Martin Werner[10], Kei Yoshimura[11], and Kira Rehfeld[1,12]

[1]Institute of Environmental Physics, Heidelberg University, Heidelberg, D-69120, Germany
[2]Department of Physical Geography and Bolin Centre for Climate Research, Stockholm University, Stockholm, SE-106 91, Sweden
[3]Department of Chemistry, Biochemistry and Pharmaceutical Sciences and Oeschger Centre for Climate Change Research, University of Bern, 3012 Bern, Switzerland
[4]Potsdam Institute for Climate Impact Research, Telegrafenberg, 14473 Potsdam, Germany
[5]GFZ German Research Centre for Geosciences, Section "Climate Dynamics and Landscape Development", 14473 Potsdam, Germany
[6]NASA Goddard Institute for Space Studies and Center for Climate Systems Research, Columbia University, New York, USA
[7]Department of Atmospheric Sciences, Cochin University of Science and Technology, India
[8]Earth Research Institute, University of California Santa Barbara, CA, USA
[9]Department of Geology, Lund University, Lund, SE-223 62, Sweden
[10]Alfred Wegener Institute, Helmholtz-Centre for Polar and Marine Research, Bremerhaven, D-27515, Germany
[11]University of Tokyo, Tokyo, Japan
[12]Geo- und Umweltforschungszentrum, Schnarrenbergstr. 94-96, 72074 Tübingen, Germany

★These authors contributed equally to this work
Correspondence to: J. Bühler (jbuehler@iup.uni-heidelberg.de) and J. Axelsson (josefine.axelsson@natgeo.su.se)

**Abstract.** The incorporation of water isotopologues into the hydrology of general circulation models (GCMs) facilitates the comparison between modelled and measured proxy data in paleoclimate archives. However, the variability and drivers of measured and modelled water isotopologues, and indeed the diversity of their representation in different models are not well constrained. Improving our understanding of this variability in past and present climates will help to better constrain

future climate change projections and decrease their range of uncertainty. Speleothems are a precisely datable paleoclimate archive and provide well preserved (semi-)continuous multivariate isotope time series in the lower and mid-latitudes, and are, therefore, well suited to assess climate and isotope variability on decadal and longer timescales. However, the relationship between speleothem oxygen and carbon isotopes to climate variables also depends on site-specific parameters, and their comparison to GCMs is not always straightforward.

Here we compare speleothem oxygen and carbon isotopic signatures from the Speleothem Isotopes Synthesis and AnaLysis database version 2 (SISALv2) to the output of five different water-isotope-enabled GCMs (ECHAM5-wiso, GISS-E2-R, iCESM, iHadCM3, and isoGSM) over the last millennium (850–1850 common era, CE). We systematically evaluate differences and commonalities between the standardized model simulation outputs. The goal is to distinguish climatic drivers of variability for both modelled and measured isotopes.

We find strong regional differences in the oxygen isotope signatures between models that can partly be attributed to differences in modelled temperatures. At low latitudes, precipitation amount is the dominant driver for water isotope variability, however, at cave locations the agreement between modelled temperature variability is higher than for precipitation variability. While modelled isotopic signatures at cave locations exhibited extreme events coinciding with changes in volcanic and solar forcing, such fingerprints are not apparent in the speleothem isotopes, and may be attributed to the lower temporal resolution





of speleothem records compared to the events that are to be detected. Using spectral analysis, we can show that all models underestimate decadal and longer variability compared to speleothems, although to varying extent.

We found that no model excels in all analyzed comparisons, although some perform better than the others in either mean or variability. Therefore, we advise a multi-model approach, whenever comparing proxy data to modelled data. Considering

karst and cave internal processes through e.g. isotope-enabled karst models may alter the variability in speleothem isotopes and play an important role in determining the most appropriate model. By exploring new ways of analyzing the relationship between the oxygen and carbon isotopes, their variability, and co-variability across timescales, we provide methods that may serve as a baseline for future studies with different models using e.g. different isotopes, different climate archives, or time periods.

**1 Introduction**

Under the current anthropogenic warming trend (Shukla et al., 2019), the interest in understanding its impacts on the mean temperature and precipitation, and changes in their variability increases. Evaluating the representation of the mean state as well as variability of past climate as simulated by climate models is crucial for reliable future projections (Schmidt et al., 2012).

Natural variability in the composition of stable water isotopes (SWI), i.e the ratio between $H_2^{16}O$ and its heavier iso-topologues $H_2^{18}O$ and HDO, constitutes an effective tracer of the water cycle and atmospheric processes. Oxygen isotope composition can be measured from many paleoclimate proxy archives such as trees, ice cores, corals, or marine and lake sediments, which collectively extend our knowledge of climatic change beyond the instrumental record (Bradley, 1999). They are usually given in the $\delta$-notation as $\delta^{18}O = \left( \frac{\frac{^{18}O}{^{16}O}_{sample}}{\frac{^{18}O}{^{16}O}_{standard}} - 1 \right) \cdot 1000\ ‰$, against the Vienna Standard Mean Ocean Water

(V-SMOW, Dansgaard, 1964; Kendall and Caldwell, 1998), while for carbonate the standard is Vienna Pee Dee Belemnite (V-PDB, Craig, 1957).

Speleothems are secondary cave deposits, which form in karst systems globally, most commonly in the low- to mid-latitudes, under a wide range of climate conditions providing precisely and absolutely dated, (semi-)continuous time series of proxy data (Wong and Breecker, 2015; Comas-Bru et al., 2019). Oxygen and carbon isotopes ($\delta^{13}C$) are incorporated in

calcite or aragonite matrices in accumulated growth layers and have long been used as proxies of terrestrial climate (Hendy, 1971).

Broad correspondence between speleothem $\delta^{18}O$ and surface temperature (e.g. McDermott et al., 2001) or local rainfall strength and seasonality (e.g. Medina-Elizalde et al., 2016; Kennett et al., 2012; Cheng et al., 2016) and between speleothem $\delta^{13}C$ and vegetation cover can be resolved in global analyses (Comas-Bru et al., 2019; Fohlmeister et al., 2020; Baker

et al., 2019; Lechleitner et al., 2021). Modification of these signatures by vadose-zone fractionation (Tremaine et al., 2011; Grossman and Ku, 1986; Romanek et al., 1992), karst hydrology, and internal cave conditions (Fairchild and Baker, 2012; Wackerbarth et al., 2010; Jean-Baptiste et al., 2019; Fohlmeister et al., 2020), and differences in geochronological methods between records, can complicate paleoclimatic interpretations (Breitenbach et al., 2012; Rehfeld and Kurths, 2014). Age-model standardization (Comas-Bru et al., 2020b), multiproxy approaches (Tremaine and Froelich, 2013; Warken et al.,

2018), and cave microclimate and dripwater chemistry monitoring (Baker et al., 2014; Treble et al., 2015), however, allow for statistically robust time-series comparisons and substantially improve our ability to disentangle climatic influences from site-specific processes across disparate climate zones (Fohlmeister et al., 2017).

Depending on the specific site, speleothem carbon isotopes can be easier to interpret than oxygen isotopes (Scholz et al., 2012; Ridley et al., 2015), especially during large climate changes such as the deglaciation (Genty et al., 2006). Studies

considering both isotopes profited from the isotopes' mutual information on fractionation processes and were able to disen-tangle the encoded climatic signal (Fohlmeister et al., 2017; Baker et al., 2017; Novello et al., 2019). In these studies, oxygen





and carbon are analyzed as proxies for different climate variables, e.g. $\delta^{18}O_{speleo}$ for Indian Summer Monsoon strength and $\delta^{13}C_{speleo}$ for local hydro-climate (Lechleitner et al., 2017; Novello et al., 2021) or $\delta^{18}O_{speleo}$ for temperature and $\delta^{13}C_{speleo}$ for C3 to C4 change in vegetation type (Dorale et al., 1992; Voarintsoa et al., 2017).

Incorporating SWI within the Earth's hydrological cycle in atmospheric general circulation models (AGCMs), general
circulation models (GCMs), and the most complex Earth system models (ESMs) is usually done by adding an additional water cycle to the hydrology of the model under explicit consideration of isotopic fractionation processes through water phase changes (e.g. Tindall et al., 2009; Yoshimura et al., 2008; Werner et al., 2016; Brady et al., 2019; Lewis and Legrande, 2015). This opens new possibilities to study and analyze past and present climates and to compare modelled climate to the archived isotopic signatures (for example, Werner, 2010; Sturm et al., 2010; Xi, 2014).

The Speleothem Isotope Synthesis and Analyses (SISAL) working group has collected a large number of speleothem records globally and compiled the database SISALv2. It has been employed for model-data comparisons of the last glacial maximum, the Mid-Holocene, the last millennium, and the historical period using different models (iCESM: Midhun et al. (2021), iHadCM3: Bühler et al. (2021), ECHAM5-wiso: Comas-Bru et al. (2019); Parker et al. (2021) and GISS-E1-R: Parker et al. (2021)), supporting the usage of the database to evaluate modelled $\delta^{18}O$ across different time periods, as the
method reproduces first-order spatial patterns of isotopic variability (Comas-Bru et al., 2019).

Extensive multi-model comparisons exist for past, present and future as the Paleoclimate Model Intercomparison Project (PMIP3/PMIP4 Jungclaus et al., 2010; Kageyama et al., 2018) under the overarching Climate Model Intercomparison Project (CMIP5/CMIP6 Taylor et al., 2012; Eyring et al., 2016) to better understand the causes of model spreads in future projections. Comparisons between models are abundant (Shi et al., 2016; Ba et al., 2014), especially for temperature and precipitation
(Parsons et al., 2020; Seftigen et al., 2017), and the impact of external forcing has been studied intensively (Atwood et al., 2016; PAGESHydro2k-Consortium, 2017). Simulations of the historical period (1850-2014CE as in Eyring et al. (2016)), or the last millennium (850-1850CE as in Eyring et al. (2016)) that will be the focus of this study, as well as idealized experiments under a range of natural and external forcings are evaluated under different variables. Water isotopes, however, have not been included in the CMIP5/CMIP6 assessments (Taylor et al., 2012; Eyring et al., 2016).

Due to this lack of systematic intercomparison and assessments with and between SWI model simulations, the Stable Water Isotope Intercomparison Group (SWING) was formed. SWING compares isotope-enabled model simulations with observations over the historical period and provides a large dataset to the scientific community (Risi et al., 2012b). The second evaluation in the SWING2-intercomparison of isotope-enabled AGCMs in 2012 showed that model differences most likely arise from differences in processes that control atmospheric humidity (Risi et al., 2012a). Conroy et al. (2013) found
that models which realistically capture precipitation patterns in the tropics are not necessarily successful in simulating the isotopic composition of precipitation compared to measured data and vice versa, cautioning on always using multiple models when comparing to paleoclimate proxy records.

All models that are used in this study have been part of the SWING2 assessment for the historical period in their current, previous, or atmosphere-only version. Therefore, this multi-model comparison complements previous work (Jungclaus et al.,
2017; Midhun and Ramesh, 2016; Conroy et al., 2013), through its focus on the representation of SWI in different models over the entire last millennium. We aim to identify common model biases (Kageyama et al., 2018) globally and in different regions, as well as distinguish specific climate drivers for modelled isotope variability on decadal and longer timescales.

Variability in models can be either internal resulting from internal interactions and processes, or external as a consequence of changes in radiative forcings (e.g. GHG, volcanoes, and solar irradiance as in Fig. 1). Variability in the speleothem
isotopic signal can also be a consequence of external climate-related variability as reflected in climate modes (e.g. El-Niño Southern Oscillation (Sun et al., 2018; Midhun et al., 2021), the North Atlantic Oscillation (Scholz et al., 2012) or the Indian summer monsoon (Fleitmann et al., 2007; Neff et al., 2001)) or changes in radiative forcing. Variations in $\delta^{18}O$ and $\delta^{13}C$ are commonly attributed to changes in solar radiation as a consequence of its influence on climate modes of variability,



temperature or precipitation (Warken et al., 2021; Lone et al., 2014; Cosford et al., 2008; Neff et al., 2001). While modelled variability commonly underestimates measured variability in paleoclimate archives with increasing discrepancies on longer timescales (Laepple and Huybers, 2014a), internal variability in speleothems may also overlay the archived signal. Especially on subdecadal to decadal timescales, lag time between the surface rainfall and the cave drip water as well as the usually slow

response of the cave micro-climate to the surface climate dampens the signal.

The last millennium is a suitable time period for model-data comparisons, as it provides an opportunity to study variability on decadal and longer timescales and to decipher internal variability from externally forced variability (Kageyama et al., 2018). Boundary conditions such as orbital forcing, sea level, and ice sheets are close to present-day, and external variability is mostly driven by variations in volcanic eruptions (Schurer et al., 2014; Neukom et al., 2019; Legrande and Anchukaitis,

2015). It is a key-paleoclimate period for the CMIP5 and CMIP6 experiments (Taylor et al., 2012; Eyring et al., 2016) and speleothem records are abundant in this period (Bühler et al., 2021).

Here we will present a multi-model comparison of five isotope-enabled last millennium simulations: ECHAM5/MPI-OM (Sjolte et al., 2018), GISS ModelE2-R (Lewis and Legrande, 2015; Colose et al., 2016a, b), the iGCM version of the Community Earth System Model (CESM) (Stevenson et al., 2019; Brady et al., 2019), the iGCM version 3 of the Hadley

Model (HadCM) (Bühler et al., 2021), and the water isotope-incorporated Scripps Experimental Climate Prediction Center's GSM (Yoshimura et al., 2008), with climate characteristics and forcings as depicted in Fig. 1 and listed in Tab. 1.

With this study, we aim to contribute to the understanding of both model and data: how do different simulations model oxygen isotopes in the hydrological cycle and how do they compare to archived speleothem data as well as what processes influence speleothem isotope composition and what effects of variability can be captured and later analyzed. We first compare

their similarities and differences in the isotopic signatures of precipitation globally (Sect. 4.1) as well as particularly at the cave site locations of a large number of high-resolution speleothems from the SISALv2 database (Comas-Bru et al., 2020b). Through spatial testing of climate variables, we analyze the relationship between the measured stable isotopes of oxygen and carbon and different modelled climate variables (Sect. 4.2). In a second step, we investigate if models can reproduce the variability as measured in speleothems on annual to centennial timescales (Sect. 4.3). Finally, we test what timescales of

events, e.g. volcanic eruptions or variations in the solar irradiance, speleothems archives are able to resolve through either isotope (Sect. 4.4).

## 2    Data

In this study, we collected and standardized the output from five different isotope-enabled model simulations over the last millennium, as well as oxygen and carbon isotopes in speleothems from the SISALv2 database.

### 2.1    Isotope-enabled general circulation models

A major advantage given by modelling SWI is its ability to both temporally and spatially resolve the variability of isotopes in precipitation by adding $H_2^{18}O$ and HDO to the part which already simulates and traces the most abundant water isotope, $H_2^{16}O$. Simulated $\delta^{18}O$ will further be denoted as $\delta^{18}O_{sim}$. In the atmospheric advection scheme of the model, which is generally a part of the model's dynamical core, all three water isotopologues behave identically. Whenever there is a phase change

(such as melting, condensing, evaporating, and freezing), additional fractionation effects are applied to the two less abundant water isotopologues. These phase changes typically occur in evaporation from land and ocean surfaces, condensation during formation of clouds, rain, snow, and re-evaporation of precipitation below cloud (for example, Werner, 2010; Sturm et al., 2010).

The models used in this study range from AGCMs forced with SST and sea-ice distribution to AOGCMs. Their basic

characteristics and boundary conditions are listed in Tab. 1. They are both used individually in the analysis, as well as by the



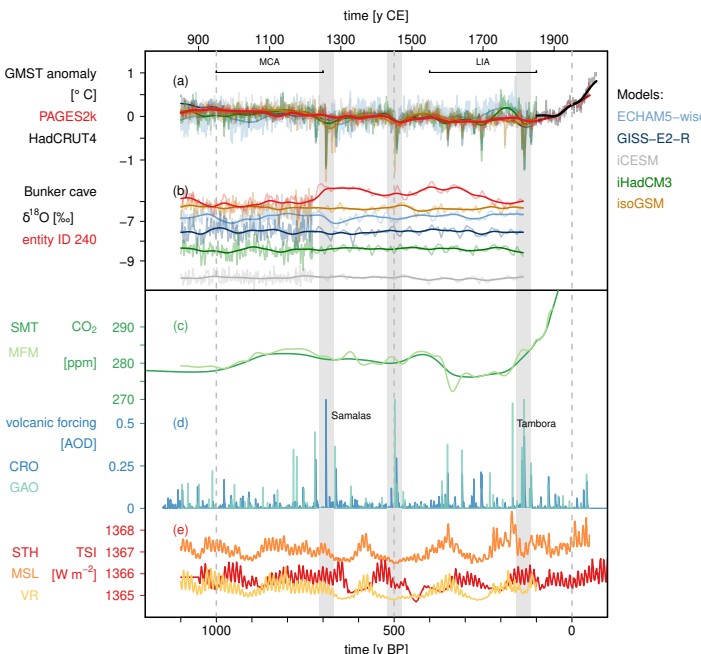

**Figure 1.** Climate as represented by the different models (ECHAM5-wiso (light blue), GISS-E2-R (dark blue), iCESM (grey), iHadCM3 (green) and isoGSM (orange)) and external forcings over the last millennium: a) global mean surface temperature anomaly as represented by the model (in model colors), as well as the reconstructed temperature anomaly (PAGES1k, red, PAGES2k-Consortium (2019)), and observed temperatures (HadCRUT4, black Morice et al. (2012)). b) isotopic composition of precipitation in the different models (in model colors) at the cave site location of Bunker cave (Germany) including the $\delta^{18}O_{speleo}$ of entity ID 240 (red), all at the temporal resolution of entity ID 240 (Comas-Bru et al., 2020b; Fohlmeister et al., 2012). c) Atmospheric $CO_2$ concentration (SMT: Schmidt et al. (2012), MFM: MacFarling Meure et al. (2006)), d) volcanic forcing in units of aerosol optical depth (AOD) (CRO: Crowley et al. (2008); Crowley and Unterman (2013), GAO: Gao et al. (2008)), where the AOD for the Gao et al. (2008) reconstruction was estimated by dividing the sulfate loading by 150 Tg (following Atwood et al., 2016), e) total solar irradiance (TSI) (STH: Steinhilber et al. (2009), MSL: Muscheler et al. (2016), VR: Vieira, L. E. A. et al. (2011)). The grey bars mark particular periods of high volcanic forcing.

ensemble mean of all models. Fig. 1 shows the climate as represented by the different models and external forcings used in the simulations.

### 2.1.1 ECHAM5/MPI-OM

We use the isotope-enabled version of the fully coupled Earth System Model ECHAM5/MPI-OM (hereafter, ECHAM5-
5    wiso) (Werner et al., 2016; Jungclaus et al., 2006). The model consists of the atmospheric model ECHAM5 (Roeckner et al., 2003) and the ocean model MPI-OM with an embedded sea-ice model (Marsland et al., 2002). The millennium-long simulation by Sjolte et al. (2018) covers the period 800-2000 CE, and uses a similar setup as the E1 COSMOS ensemble by Jungclaus et al. (2010), but with a different solar forcing based on the solar modulation record inferred from combined neutron monitor and tree-ring $^{14}C$ data (Muscheler et al., 2016).
10    Isotope diagnostics have been implemented for the atmosphere, ocean, and land surface component of the model and are computed throughout the entire water cycle in the ECHAM5 (Werner et al., 2016) and MPI-OM (Werner et al., 2016). The



**Table 1.** Basic characterization of the last millennium simulation.

|  | ECHAM5/MPI-OM | GISS ModelE2-R | iCESM1 | iHadCM3 | isoGSM |
|---|---|---|---|---|---|
| Reference | Sjolte et al. (2018); Werner et al. (2016) | Lewis and Legrande (2015); Colose et al. (2016a, b) | Brady et al. (2019); Stevenson et al. (2019) | Bühler et al. (2021); Tindall et al. (2009) | Yoshimura et al. (2008) |
| Years | 800-2005 CE | 850-1979 CE | 850–2005 CE | 850-1850 CE | 851-2000 CE |
| Atmospheric resolution | $3.75° \times 3.75°$ | $2.5° \times 2°$ | $2.5° \times 1.875°$ | $3.75° \times 2.5°$ | $1.875° \times 1.875°$ |
| Orography | fixed to 0BP | ETOPO1 fixed to 0BP | GTOPO30 fixed to 0BP | fixed to 0BP | ETOPO5 fixed to 0BP |
| Orbital Parameter | Variation Seculaires des Orbites, Planetaires (VSOP) analytical solution by Bretagnon and Francou (1988) | Berger and Loutre (1991) | Berger (1978) | fixed to 0BP | a millennium trend is considered |
| GHG | $CO_2$, $CH_4$, $NO_2$: MacFarling Meure et al. (2006) Historical, anthropogenic: Marland et al. (2003)) Ozone: Climatology of Paul et al. (1998) | Transient from 850 (Schmidt et al., 2011) | well-mixed greenhouse gases ($CO_2$, $CH_4$, $NO_2$) from high-resolution Antarctic ice cores Schmidt et al. (2011) | well mixed $CO_2$, $CH_4$, $NO_2$ and other trace gases (Schurer et al., 2014; Schmidt et al., 2012) | well-mixed greenhouse gases ($CO_2$, $CH_4$, $NO_2$) from high-resolution Antarctic ice cores Schmidt et al. (2012) |
| Vegetation | Pongratz et al. (2008) with vegetation from Jungclaus et al. (2010) | Pongratz et al. (2008) | Pongratz et al. (2008), starting 1500: Hurtt et al. (2011) | dynamic TRIFFID (Cox, 2001) | Pongratz et al. (2008), starting 1500: Hurtt et al. (2011) |
| Volcanic forcing | Crowley et al. (2008) | Crowley et al. (2008) | Gao et al. (2008) | Crowley and Unterman (2013) | Gao et al. (2008) |
| Total Solar Irradiance | Muscheler et al. (2016, 2007) | Steinhilber et al. (2009), starting 1850: Wang et al. (2005) | Vieira, L. E. A. et al. (2011) with 11-year cycle added similar to Schmidt et al. (2011) | Steinhilber et al. (2009); Wang et al. (2005); Schurer et al. (2014) | Vieira, L. E. A. et al. (2011), starting 1834: Lean (2009), with 11-year cycle added from Schmidt et al. (2012) |

land surface model assumes no fractionation in most of the physical processes (Haese et al., 2013). Water tracers are fully mixed and advected in the ocean model, and its total mass is conserved (Werner et al., 2016).

ECHAM5-wiso has been used extensively within the paleoclimate field, as well as for present time (for example, Werner et al., 2016; Langebroek et al., 2011; Goursaud et al., 2018). The fully coupled version of the model ECHAM5/MPI-OM

5 ESM has very good agreement with both present-day isotope observations from the GNIP database, as well as with ice core and speleothem proxies during mid-Holocene (MH, Comas-Bru et al., 2019), last glacial maximum (LGM Werner et al., 2016; Comas-Bru et al., 2019), and for last interglacial (Parker et al., 2021). Both in the ESM and with the atmospheric component (ECHAM5-wiso), a warm bias in the model is found over high-latitudinal regions, especially over Greenland and Antarctica (Werner et al., 2011, 2016) and has been attributed to the coarse spatial resolution in the atmospheric component

10 of the model (Werner et al., 2016) resulting in an underestimation of isotope depletion in these regions. The last millennium simulation has not been evaluated globally, but climate reconstructions and isotope variability have been studied in the North Atlantic region, where the amplitude of the variability was underestimated in the model compared to ice cores (Sjolte et al., 2018). Previous studies also stress the isotopic response to volcanic eruptions and phases of NAO (Guðlaugsdóttir et al., 2018, 2019).



### 2.1.2 GISS ModelE2-R

The isotope-enabled AOGCM GISS ModelE2-R (hereafter, GISS-E2-R) (Schmidt et al., 2006, 2014) is used with the same physics as in the CMIP5 experiments (Miller et al., 2014; Schmidt et al., 2014). Water tracers and isotopes are incorporated into the atmosphere, land surface, ocean, and sea-ice components of the model (Schmidt et al., 2005). Several experiments

have been set up for the last millennium with GISS-E2-R, due to uncertainties in past forcings and its effects, with different combinations of solar, volcanic, and land-use/vegetation forcings but all with the same greenhouse gas and orbital change (Colose et al., 2016a, b; Lewis and Legrande, 2015).

GISS-E2-R has been shown to simulate modern isotopic observations well, except over Antarctica, in terms of changes in convection, clouds, and isotope kinetics (Schmidt et al., 2005). For the last millennium, GISS-E2-R has also explored the

isotopic responses to volcanic eruptions in South America (Colose et al., 2016a), volcanic forcing in relation to the position of the intertropical convergence zone (Colose et al., 2016b), and to ENSO (Lewis and Legrande, 2015). The model has been shown to have a warm SST bias and issues in sea ice concentration around Antarctica, related to the transport of the Antarctic Circumpolar Current. In the tropics, a warm bias is also found over land, together with cooler northern midlatitudes (Schmidt et al., 2014).

### 15 2.1.3 iCESM1

We use the last millennium run of the isotope-enabled iCESM1 version 1.2 model (hereafter, iCESM) (Hurrell et al., 2013; Brady et al., 2019; Midhun et al., 2021), a fully-forced simulation out of an eight-member ensemble of different external forcings. As the model is open-source and publicly available, it is widely used in the scientific community, and simulations for past (Zhu et al., 2017; Zhang et al., 2012) and present climate exist (Otto-Bliesner et al., 2016).

The model consists of the isotope-enabled Community Atmosphere Model version 5.3 (iCAM5.3, isotope-enabled version based on Neale et al., 2010), a Land Model CLM4 (Oleson et al., 2010), a sea-ice model, and an ocean component that is based on the isotope-enabled POP2 (Zhang et al., 2017). Isotopes in the water cycle are represented as a new parallel hydrological cycle in all hydrological components in the atmosphere, ocean, land, and sea ice in the form of numerical water tracers and can be tracked in space and time.

The isotope-enabled version captures general global isotopic signatures well over ocean areas but shows small discrepancies across the land surface (Brady et al., 2019). This effect has been explained by the model showing a slight negative isotopic bias due to overestimated modelled convection in mid-latitude oceans. Consequently, the transport of SWI-mass poleward and landward has been deemed insufficient (Nusbaumer et al., 2017). Footprints associated with major climatic modes such as ENSO and PDO are found to be well represented also in isotopic signatures (Midhun et al., 2021).

### 30 2.1.4 iHadCM3

We use the last millennium run from the fully coupled atmosphere-ocean isotope-enabled GCM iHadCM3 (Bühler et al., 2021). iHadCM3 has been widely used to simulate present (Dalaiden et al., 2020) and future climate (Sime et al., 2008; Tindall et al., 2009; IPCC, 2013), as well as for past climates (Tindall et al., 2010; Sime et al., 2013; Holloway et al., 2018). The model consists of several components: the atmosphere model HadAM3 (Pope et al., 2000), the ocean model HadOM3

(Gordon et al., 2000), a sea ice model (as described in Valdes et al., 2017) and a dynamic land surface and vegetation model (Cox, 2001).

For the isotope-enabled version, SWI were added as two separate water species in the atmospheric model, and as tracers in the ocean model. Fixed isotope fractions are added to a fixed volume gridbox of the ocean and experience changes due to evaporation, precipitation, and runoff through a virtual isotope flux, altering the $\delta^{18}O_{sim}$ ratio in the top level of the ocean

accordingly (Tindall et al., 2009).



Compared to instrumental observations, the model represents sea surface temperature, ice sheet, and ocean heat content well (Gordon et al., 2000). The freshwater hydrological cycle in the model shows only a slight overestimation in the local evaporation (Pardaens et al., 2003). The model simulates the major isotopic fractionation effects defined by Dansgaard (1964) (e.g. the latitude effect, the amount effect, and the continental effect) appropriately compared to GNIP data (Zhang et al., 2012). Additionally, a broad agreement in isotopic output with GNIP data in the general spatial distribution can be observed (Tindall et al., 2009).

### 2.1.5 isoGSM

IsoGSM is the isotope-enabled version of the Scripps Experimental Climate Prediction Center's (ECPC) GSM (hereafter, isoGSM) (Yoshimura et al., 2008). The model is based on the previous medium-range forecast model used at NCEP, making it well documented in its performance as an operational weather forecast model (Kanamitsu et al., 2002; Caplan et al., 1997).

The IsoGSM is a stand-alone atmospheric model. Here, it has been forced with SST and sea-ice distributions from CCSM4 last millennium simulation (Landrum et al., 2013). Land surface processes are modelled through NOAH model, but isotopic fractionation is not considered in these processes (Yoshimura et al., 2008).

isoGSM has been shown to represent isotope and precipitation observations globally using a spectral nudging technique captured by the NCEP/DOE Reanalysis dataset (Yoshimura et al., 2008). Its last millennium simulation has not been evaluated in previous studies, but isoGSM captures large-scale isotope and climate patterns in present times compared to other models with implemented isotopes (Risi et al., 2012b). The model has shown to also reproduce observed isotopic and precipitation variability well over the regions of Indian Summer Monsoon (Berkelhammer et al., 2012a), western North America (Berkelhammer et al., 2012b), and NW Scotland (Baker et al., 2012). Recently, IsoGSM showed good consistency with speleothem oxygen isotopes from East Asia (Chiang et al., 2020) and from South Asia (Kathayat et al., 2021).

isoGSM tends to underestimate the depletion of $\delta^{18}O_{sim}$ in dry regions such as the continent of Antarctica (Yoshimura et al., 2008). A decreasing summer temperature increases the precipitation $\delta^{18}O_{sim}$, and is caused by the moisture transport scheme of the model associated with areas of extremely dry conditions (Yoshimura et al., 2008; Risi et al., 2012b).

### 2.2 SISALv2 database

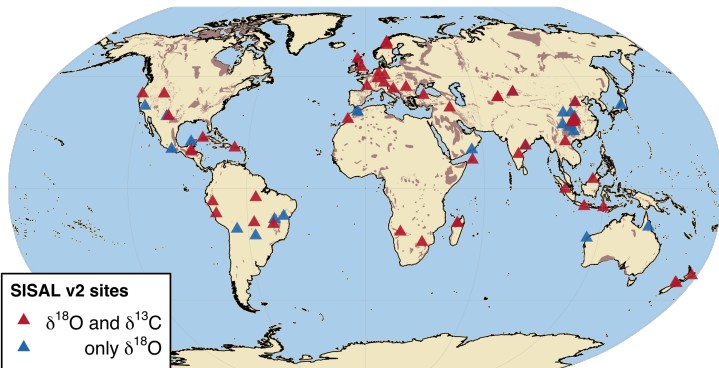

**Figure 2.** Site locations of the SISALv2 database on a global karst map (brown shading Williams and Ford (2006). We only consider entities that cover a minimum of 600 year period within the last millennium and that include at least 2 dates and 36 isotopic measurements in this period. Shown are all sites within the last millennium meeting these selection criteria that have both oxygen and carbon isotope measurements (red), and only oxygen isotopes measurements (blue).





In this study, we use speleothem data from the Speleothem Isotope Synthesis and Analysis version 2 (SISALv2) database (Comas-Bru et al., 2020a, b). The database includes 691 speleothem records from 294 caves across the globe, from all continents except Antarctica.

The last millennium has abundant records globally with sufficient resolution and reasonable dating uncertainties (Bühler
et al., 2021). We filtered the database for records that cover at least a 600 yr period within the last millennium (850-1850CE), exhibit at least two dates within the time period, as well as 36 stable isotope measurements, to guarantee a minimum resolution of 30 yr. We obtain 89 records from 75 different sites for $\delta^{18}O_{speleo}$ of which 58 (65 %) from 50 sites also have $\delta^{13}C_{speleo}$ measurements (Fig. 2).

## 3   Methods

To compare climate simulation outputs and speleothem data, both need some preparation. Modelled output comes in regular monthly resolution, while time series of speleothem proxies are irregularly sampled. Different speleothem mineralogies, as well as different isotopic standards between modelled and recorded data, need to be accounted for before statistical similarity measures are applied.

### 3.1   Speleothem drip water conversion

The database includes calcite, aragonite, and mixed mineralogy speleothem records. Following Comas-Bru et al. (2020b), we only use pure calcite or aragonite speleothems. To be able to compare precipitation $\delta^{18}O_{sim}$ values to those measured in calcite or aragonite, the $\delta^{18}O_{speleo}$ is converted to its *drip water equivalent* ($\delta^{18}O_{dweq}$) relative to the V-SMOW standard as in Comas-Bru et al. (2019). This conversion is temperature dependent but to different extents for both minerals.

Tremaine et al. (2011) provide an empirically based fractionation formula for speleothems of calcite mineralogy:

$$\delta^{18}O_{dweq} = \delta^{18}O_{calcite} - \left(\left(\frac{16.1 \cdot 1000}{T}\right) - 24.6\right), \tag{1}$$

where $T$ is the temperature in K and $\delta^{18}O$ are given in units of ‰.

Aragonite speleothems form under different conditions, e.g. higher Mg content of the dripping solution or very slow drip-rate (Fairchild and Baker, 2012), resulting in a different fractionation factor compared to calcite as described by Grossman and Ku (1986):

$$\delta^{18}O_{dweq} = \delta^{18}O_{aragonite} - \left(\left(\frac{18.34 \cdot 1000}{T}\right) - 31.954\right). \tag{2}$$

For both conversions, the cave temperature at the time of the fractionation is needed. As these are often not available, especially in a palaeoclimate setting, we use annual mean modelled surface temperatures as a surrogate for cave temperatures (Fairchild and Baker, 2012). For caves in very cold conditions, the annual mean surface temperature may underestimate the mean cave temperature by some degrees due to long-lasting snow-packs. This underestimation, however, only corresponds to
an overestimation of 1 ‰ in $\delta^{18}O_{dweq}$ and is within the range of the simulation ensemble. Additionally, cave-dependent time lags between the surface and the cave temperature are not accounted for, as they have a negligible effect on the time-averaged mean isotopic value. The conversion is done for each entity and each simulation individually, where we use the simulated annual mean surface temperature, down-sampled to the record's resolution.

In a last step, the V-PDB values are converted to V-SMOW, using the conversion from Coplen et al. (1983):

$$\delta^{18}O_{SMOW} = 1.03092 \cdot \delta^{18}O_{PDB} + 30.92. \tag{3}$$





For carbon isotopes, different fractionation paths exist depending on mineralogy. Following Fohlmeister et al. (2020), we convert the aragonite $\delta^{13}$C values to corresponding calcite values using a fractionation offset of $1.9 \pm 0.3$ ‰ ($\delta^{13}\text{C}_c = \delta^{13}\text{C}_{calcite} = \delta^{13}\text{C}_{arag} - 1.9 \pm 0.3$ ‰). This offset accounts for the different enrichment-factors of the two polymorphs as established in laboratory studies (Romanek et al., 1992) and confirmed in a speleothem study (Fohlmeister et al., 2018).

From the drip water conversion, we obtain a matrix for each of the isotopes with one row per measurement and six columns, where one represents the observations and the other five the modelled estimates.

### 3.2   Data processing

Although all simulation output from the five different climate models is available at monthly resolution, the time coverage differs. All simulation runs are cut to cover the time period from 850CE to 1850CE, an interval that is similar to PMIP's

interval in the last millennium experiment. All simulation runs provide different sets of post-processed output variables. We focus on surface temperature, precipitation, precipitation-$\delta^{18}\text{O}_{sim}$, and evaporation. For the simulations lacking evaporation as a diagnostic (iCESM, isoGSM, and iHadCM3), we convert latent heat to potential evaporation and use these variables within the simulations. Outliers in the simulation are removed by comparing each modelled value in the 3D-output data matrix with its eight neighbors in time and space. If the value deviates from the mean of these eight values by more than five

standard deviations, the value is set to NA. On average 0.001% of values are set to NA through this method.

    Monthly mean $\delta^{18}\text{O}_{sim}$ are used with weighting by precipitation minus evaporation amount (infiltration adjusted precipitation weighting, $iw$) to obtain annual values. The simulated isotope mean is calculated as:

$$\delta^{18}\text{O}_{\text{iw}} = \frac{\sum_1^n \delta^{18}O_k iw_k}{\sum_1^n iw_k} \tag{4}$$

where $\delta^{18}\text{O}_{iw}$ is the monthly weighted annual ~~annually weighted~~ composition of isotopes, $\delta^{18}O_k$ refers to monthly sim-

ulated $\delta^{18}$O, and $iw_k$ is the corresponding monthly amount of $iw$. As isotopic fractionation occurs during evaporation from the soil, models, where $\delta^{18}\text{O}_{sim}$ is also available for soil layers, would be more realistic to compare to speleothem data. However, these were only available for a few simulations. Using infiltration-weighted $\delta^{18}\text{O}_{sim}$, therefore, offered a more equal handling of the data and enabled a better comparison of results.

    Where simulation data are compared on a gridbox-level, we block-average all simulations to the same spatial resolution

as that of the ECHAM5-wiso-run, which has the lowest spatial resolution. Data at the speleothem cave sites are extracted via bi-linear interpolation as in Bühler et al. (2021). Here, a gridbox of the size of the simulation's original resolution with the cave's location at its center is resampled from the original gridboxes that it overlaps with.

    The simulated monthly temperature, precipitation, and evaporation at the speleothem cave locations are averaged to annual mean values. All speleothems in our last millennium subset, however, come as irregularly sampled time series with

a median resolution of 5.19 yr per entity, 90% CI: (4.13, 6.99). Considering only the speleothems with measurements of both isotopes yields a median resolution of 6.08 yr (4.07, 7.85). All simulated variables of all models are block-averaged to the irregular temporal resolution of the individual speleothem. We also include speleothems in our analysis where no $\delta^{13}$C measurements but only $\delta^{18}$O are available. In direct comparisons between carbon and oxygen isotopes, we only consider those 58 speleothems that provide samples for both isotopes.

The relationship between $\delta^{18}\text{O}_{speleo}$ and simulated climate variables are determined following three different latitude bands to guarantee enough data points within each zone, the tropics, the subtropics (poleward of the subtropical jet stream), and the extratropics (Holden, 2012). The tropics are commonly defined as the zone between the Tropic of Cancer and the Tropic of Capricorn (23.44°S to 23.44°N); the subtropical region 23.44 to 35°N/S, and the extratropical region 35 to 90°N/S (with cave sites only extending to 66°N and 42°S).





Spatial testing between speleothem $\delta^{18}O_{dweq}$, $\delta^{13}C_c$, simulated $\delta^{18}O_{sim}$, and meteorological variables is done by linear regression of the simulated millennium mean, down-sampled to the temporal resolution of each record, and entity mean. To account for the spread between simulated variables and calculated $\delta^{18}O_{dweq}$, the linear regression is done via bootstraping ($n = 2000$). Confidence intervals for all entity mean variables are also calculated via bootstrapping with a significance level

of $\alpha = 0.1$. p-values are calculated through a fit linear regression model (`fitlm.m` (MATLAB, 2018)) using Pearson's product moment correlation.

*Correlation* estimates and p-values for regular time series i.e. the annual resolution output of the simulation, are calculated via the Pearson's product moment correlation (via the function `cor.test` (R Core Team, 2020b)). We use a significance

level of $\alpha = 0.1$.

Correlation estimates for irregular time series are calculated via Pearson correlation as adapted by Rehfeld and Kurths (2014) and tested for last millennium speleothem records in Bühler et al. (2021). Here, we also choose a significance level of $\alpha = 0.1$. This level is appropriate for both the regular and irregular time series, considering the number of samples $N$ compared to the strictness and expected level of false positives. Whenever calculating correlation estimates where speleothem

data is involved, we use the raw $\delta^{18}O_{speleo}$ or $\delta^{13}C_{speleo}$ instead of the drip-water converted values to decrease any potential biases.

With all simulated variables down-sampled to the irregular resolution of each speleothem record, the use of power spectral analysis of the time series can describe the variation of common signals on a frequency spectrum of all time series (Chatfield,

2003). The *power spectral density* (PSD) of a time series describes the power distribution versus frequency over a finite interval of time (Chatfield, 2003).

To compute spectra of irregularly sampled time series, these are first interpolated to their mean resolution by a double interpolation and filtering process (following Laepple and Huybers, 2014a; Rehfeld et al., 2018; Dolman et al., 2020). This interpolation is performed to reduce high-frequency artifacts. The robustness of this spectral estimation process was recently

confirmed by Hébert et al. (2021).

### 3.3 Synchronous events in speleothem isotopes

An alternative similarity measure to correlation estimates, especially given the irregularity of the available time series, is checking for synchronous events within two time series. After distinguishing extreme events, strength and direction of synchronous extreme events are only based on their relative timing (Rehfeld and Kurths, 2014). In this study we focus only on

this relative timing.

Within the modelled or measured isotope time series, we distinguish the $5\%$ extreme values as the values above/below the $97.5\%$ / $2.5\%$ quantile of the time series distribution. Two extreme values occurring in a row are treated as two separate extreme events. Extreme values for time series of solar irradiance are determined in the same manner. For the volcanic forcing time series, we distinguish those events above the $95\%$ quantile of the distribution.

Two events are considered synchronous, if they both occur within a time period around the events, limited by a local threshold $\tau$. This local threshold is calculated for each possible pair of extreme events and is chosen as half the minimum time between either extreme event and its preceding or succeeding extreme. The median $\tau$ is $4.62$ yr (90% CI: $4.37$, $5.28$), which is of the same order of magnitude as the median resolution of all records with both carbon and oxygen measurements of $6.08$ yr, ($4.07$, $7.85$). We set a hard threshold limit of $50$ yr, corresponding to the median age uncertainty considering the original

chronologies as well as the SISALv2 ensemble chronologies (Comas-Bru et al., 2020b). When comparing synchronous





events between isotopes within one particular speleothem, age-uncertainties are negligible in the comparison as $\delta^{13}C_{speleo}$ and $\delta^{18}O_{speleo}$ values stem from the same measurement of individual sub-samples.

For comparable extreme event signatures between the modelled and measured isotopes to volcanic or solar forcings, each modelled time series is checked for synchronicity against their respective forcing time series. The analysis is repeated for the
5 speleothems for the same number of forcings and averaged.

When looking at the temporal distribution of global synchronous extreme events, they are sorted into bins of $10\,yr$, which is approximately twice the median local $\tau$. If a pair shows several extreme events within one bin, it is only counted once. We determine significance by randomly permuting one of the time series of a pair and repeating the analysis 2000 times. Within one bin, all counts that are larger than the 95% quantile of this 'mean background noise' can be regarded as significant.

## 4 Results

### 4.1 Overview of simulated vs. measured speleothem oxygen isotopes

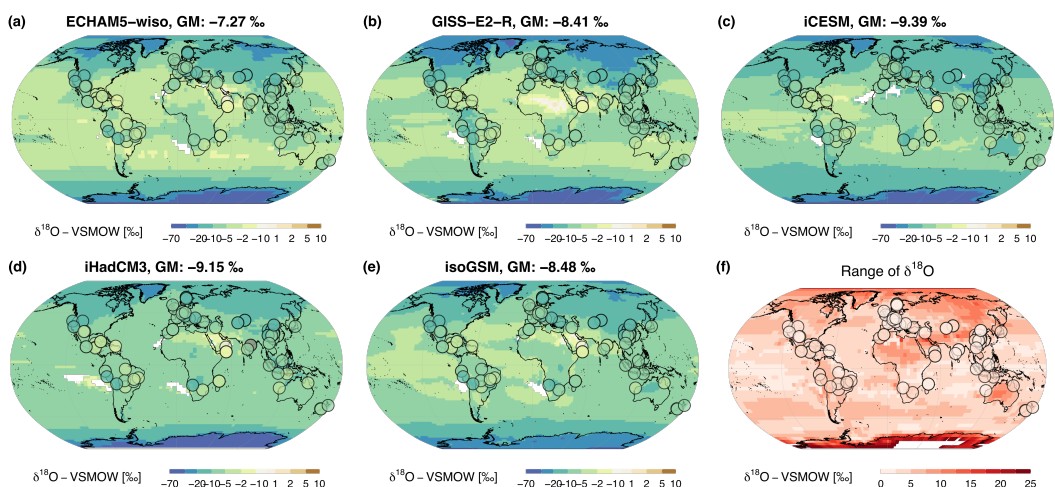

**Figure 3.** Simulated $\delta^{18}O_{iw}$ climatology (a-e) of the respective simulation: a) ECHAM5-wiso, b) GISS-E2-R, c) iCESM, d) iHadCM3, e) isoGSM) in the background. The time-averages mean $\delta^{18}O_{dweq}$ using the respective simulated temperatures are depicted as circles. Global means (GM) of $\delta^{18}O_{sim}$ are given in the title of each subplot. f) shows the range of $\delta^{18}O_{sim}$ between all simulations for each gridbox, as well as the range for the difference between simulation and record. Light colors indicate large agreement between the simulations, while darker colors mark areas, where the models differ strongly and the spread between the $\delta^{18}O_{sim}$ is larger. Antarctic $\delta^{18}O_{sim}$ ranges are up to $40‰$, highlighting the different model performance in this region (white area in f).

We first compare the mean $\delta^{18}O_{iw}$ signal of the five different last millennium simulations, to see potential model biases and large differences between the simulations (Fig. 3). The global mean $\delta^{18}O_{iw}$ values are fairly similar in area-weighted global mean of $8.48‰$ (90% CI: $-8.61, -8.36$) and $-8.41‰$ ($-8.62, -8.2$) for isoGSM and GISS-E2-R, respectively. The
15 ECHAM5-wiso run is less depleted with a global $\delta^{18}O_{sim}$ mean of $-7.27‰$ ($-7.46, -7.09$), but with clearly visible more strongly depleted mid-latitude oceans than in the other simulations. iCESM and iHadCM3 show a stronger depletion of $-9.39‰$ ($-9.51, -9.28$) and $-9.15‰$ ($-9.29, -9.01$) respectively, with stronger depletion towards the poles compared to the other simulations.



This general offset between the global mean $\delta^{18}O_{iw}$ is also visible when comparing the spread of mean values on a gridbox-level (Fig. 3f), where isotopic signatures differ the strongest in the Antarctic. Restricting the view to low- to mid-latitudes, the largest model data difference is in the area of the Sahara desert, the Arabian peninsula, and the Indian peninsula, where in particular the iHadCM3 simulation deviates strongly from the other four. This deviation could be due to the higher

than multi-model mean temperature in these areas in the iHadCM3 simulation (SFig. 3). Areas of high precipitation difference between the simulations do not coincide with areas of higher spread in isotopic composition of the precipitation (compare SFig. 1 and SFig. 4).

Also shown in Fig. 3f) is the spread between the offsets to the respective $\delta^{18}O_{dweq}$ at the cave locations. A close agreement between the models does, however, not indicate a good model-data match. It only indicates that the offsets to the converted

speleothem data are similar between the models. Highest agreement of the offsets of 2.31‰ range is obtained at Tham Duon Mai Cave in Laos (siteID 159, Wang et al., 2019), while strongest disagreement of a range of 8.14 ‰ between simulations is at Huangye Cave in China (siteID 17, Tan et al., 2011).

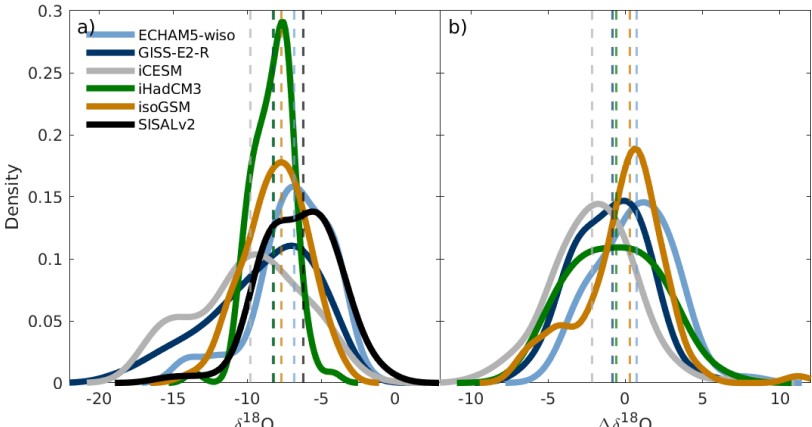

**Figure 4.** Kernel density estimates of (a) the general distrubutions in simulated and speleothem $\delta^{18}O$ at cave locations, (b) offsets between simulations and speleothems last millennium mean ($\Delta\delta^{18}O= \delta^{18}O_{iw} - \delta^{18}O_{dweq}$). Dashed lines represent the medians.

Analyzing the offsets between simulations and speleothems can indicate how well the model data matches the proxy signal. Here we investigate the offset in $\delta^{18}O$ between simulated last millennium mean ($\delta^{18}O_{iw}$) and speleothems ($\Delta\delta^{18}O= \delta^{18}O_{sim}$

- $\delta^{18}O_{dweq}$) on a global scale. The general distribution and offsets between each model and speleothem data are shown as kernel density estimates (Fig. 4). The speleothem dataset has a median general distribution of -6.21‰ globally. Of the simulations, ECHAM5-wiso has the closest distribution median with -6.82‰, followed by isoGSM (-7.72‰), iHadCM3 (-8.20‰), GISS-E2-R (-8.25‰), and iCESM (-9.79‰) (Fig. 4a). The offset distributions between simulations and speleothem $\delta^{18}O$ (Fig. 4b) are fairly symmetrical but vary between the simulations, with medians of 0.72‰, -0.86‰, -2.01‰, -0.67‰

and 0.28‰, for ECHAM5-wiso, GISS-E2-R, iCESM, iHadCM3 and isoGSM, respectively. While ECHAM5-wiso has the closest median globally, the least offset between simulation and speleothem $\delta^{18}O$ at cave locations has isoGSM, with a mean of -0.17‰ (90% CI: $-0.66, 0.33$). GISS-E2-R and iCESM have higher negative offsets, with a mean of $-1.02$‰ $(-1.41, -0.62)$ and $-2.04$‰ $(-2.50, -1.60)$, followed by iHadCM3 $-0.68$‰ $(-1.18, -0.18)$. ECHAM5-wiso is the only model that has a positive offset mean between simulation and proxy data, with a mean of $0.63$‰ $(0.20, 1.05)$, in line with the less

depleted global mean seen in Fig 3.



The largest positive offsets (less depleted $\delta^{18}O_{sim}$ than $\delta^{18}O_{dweq}$) are found at Huagapo cave in Peru (siteID 277, Kanner et al., 2013) and Minnetonka cave in the USA (siteID 200, Lundeen et al., 2013) for at least four model simulations. Highest negative offsets (at least three models agree) are found at Hoq cave on Socotra Island, Yemen (siteID 253, Van Rampelbergh et al., 2013), Diva cave in Brazil (siteID 38, Novello et al., 2012), and Qunf cave in Oman (siteID 159, Fleitmann et al.,

5      2007).

The general patterns of the isotope climatology are similar between the models (Fig. 3). Larger differences in the modelled isotopic signatures appear particularly in regions where modelled temperature spreads widely as well. On average, the offsets to the speleothem records (Fig. 4) appear small and are consistent with the general difference to precipitation $\delta^{18}O_{sim}$.

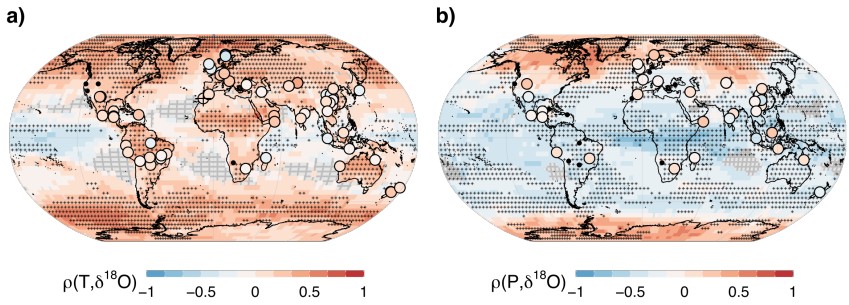

**Figure 5.** Correlations between simulated interannual SWI changes and temperature (a) and precipitation (b) in each gridbox. The background shows the mean correlation over all 5 simulations between annual $\delta^{18}O_{sim}$ and simulated annual temperature per gridbox (a), and for precipitation (b). Crosses indicate gridboxes, where correlation for four or more models has the same sign as the mean between all simulations. Symbols indicate the mean correlation of the simulated temperature (precipitation) to the recorded $\delta^{18}O_{speleo}$ at record resolution. Crossed circles mark those, where more than four models agree in the mean sign of the correlation to $\delta^{18}O_{speleo}$. Black circles indicate the location of those speleothems in the last millennium subset that show no significant correlation to any model.

Differences between $\delta^{18}O_{sim}$ signatures between the models may arise from different simulation drivers for the oxygen

isotopes, e.g temperature and precipitation, and can hint at different processes that govern the isotopic water cycle at a certain region within the simulations. The mean of the correlation to these main climatic drivers to $\delta^{18}O_{sim}$ shows high agreement between the simulations (Fig. 5, individual correlation fields in SFig. 5). For the correlation to temperature (Fig. 5a), two main domains can be distinguished: there is mainly a positive correlation to temperature in the mid- to high-latitudes and on the continents, and negative correlation in the low-latitude ocean. Large-scale agreement between the simulations is,

however, limited to the higher latitudes and the tropical ocean.

Two domains are also apparent in the correlation to precipitation (Fig. 5b), which are even more clearly separated than for temperature. We find areas of negative correlation to $\delta^{18}O_{sim}$ in the low to mid-latitudes and areas of positive correlation only in the very high latitudes. The agreement between the simulations, indicated by the crosses, is higher in the low- to mid-latitudes.

The inter-model comparison shows more agreement in the correlation fields to temperature than to precipitation, when focusing only on cave locations: the sign of correlation between $\delta^{18}O_{sim}$ and simulated temperature agree for three and more simulations at 60% of locations, and for four and more simulations even at 26% of locations. For precipitation on the other hand, only 11 % of locations agree in sign for three and more simulations, while it is only 1.1 % with agreement in four or more simulations.

The model-data comparison shows more significant temporal correlation estimates between simulated temperature to $\delta^{18}O_{speleo}$ and also more sign agreement between these correlation estimates than to simulated precipitation. For precipita-





tion, we find no cave, where more than four models agree in the sign to the mean correlation between modelled and recorded $\delta^{18}O$. For temperature, four speleothems from four different cave sites show a significant correlation of absolute strength $|c| > 0.15$ for at least four simulations.

The data suggests that two main drivers for $\delta^{18}O$ can be distinguished in specific regions - temperature is dominant in the

high latitudes, while precipitation appears to be the main driver in the low latitudes.

## 4.2 Spatial testing for climatic and environmental effects on speleothem $\delta^{18}C$ and $\delta^{13}C$

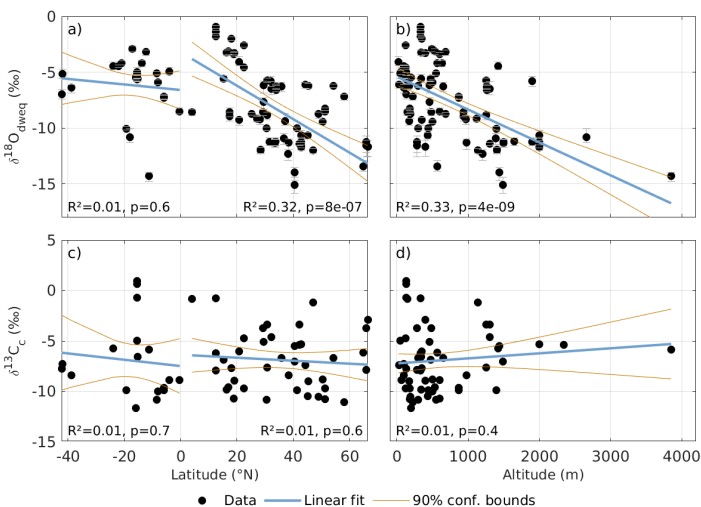

**Figure 6.** Speleothem $\delta^{18}O_{dweq}$ and $\delta^{13}C_c$ against latitude and altitude as provided by the database. Linear regression lines are shown separately for northern and southern hemisphere in (a) and (c), while the $R^2$ and p corresponds to the global linear regressions.

$\delta^{18}O$ in precipitation shows global signatures depending on latitude or altitude amongst other variables (Dansgaard, 1964). We assess this by looking at the relationships between speleothem $\delta^{18}O_{dweq}$ and $\delta^{13}C_c$ with the site-specific variables of latitude and altitude.

In Fig. 6a) we see a decrease in $\delta^{18}O_{dweq}$ as more northern speleothems are considered (globally: $R^2 = 0.22$, $p < 0.00$, STab. 1). The large spread in the mid-latitude Northern Hemisphere is mostly due to the high number of speleothem records available in both Europe and China, implying a high longitudinal spread within the database. Fig. 6b) shows a global negative relationship of $\delta^{18}O_{dweq}$ to altitude, even though records get scarcer as the altitude increases. No clear pattern is visible for $\delta^{13}C_c$ , and the dataset has a large mean spread (Fig. 6c). A statistically significant relationship between altitude and $\delta^{13}C_c$

cannot be established. However, the spread in $\delta^{13}C_c$ appears to decrease with increasing altitude (Fig. 6d)

We further explore the simulated meteorological variables and investigate spatial relationships between speleothem mean $\delta^{18}O_{speleo}$ and model variable ensemble mean (Fig. 7, STab. 1). We find a significant relationship between $\delta^{18}O_{iw}$ and speleothem $\delta^{18}O_{dweq}$ across all latitude bands (Fig. 7 a-c), with the strongest correlation in the extratropics. Furthermore, we find a global dependency of $\delta^{18}O_{dweq}$ to mean simulated temperature and precipitation at the cave sites. For both tem-

perature and precipitation, we find the strongest relationships to $\delta^{18}O_{dweq}$ in the subtropical regions. In all three regions, the relationship to temperature always exceeds that of precipitation. In Fig. 7 (d-f) cave site altitude information is applied by color codes, showing that higher altitude coincides with more negative $\delta^{18}O_{dweq}$ in the tropics ($R^2 = 0.45$, $p < 0.01$) and the extratropics ($R^2 = 0.50$, $p < 0.01$). A higher altitude also coincides with lower temperature in the same latitudinal





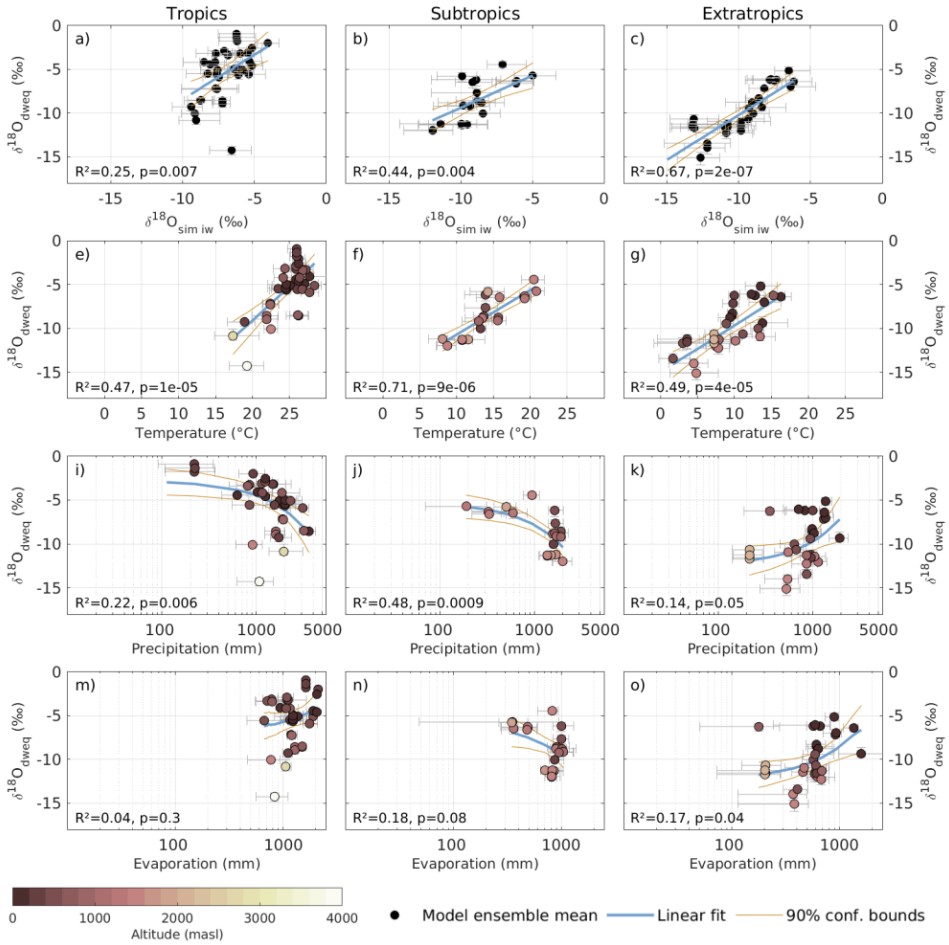

**Figure 7.** Simulated weighted $\delta^{18}O_{sim}$ (a-c), temperature (d-e), precipitation (g-i) and evaporation (j-l) against speleothem $\delta^{18}O_{dweq}$ for model ensemble mean in the tropics, subtropics and extratropics. The tropical region (23.44°S to 23.44°N) is shown in left panel (a, d, g, j); the subtropical region (23.44–35°N/S) is shown in the middle panel (b, e, h, k); the extratropical region (35–90°N/S) is shown in the right panel (c, f, i, l). In d)-l) altitude information is applied as shaded colors. We use $\delta^{18}O_{iw}$ for all simulations. Note the semi-logarithmic axes for precipitation and evaporation.

bands. For the subtropics, the cave site altitudes have less range, and an altitude pattern cannot be distinguished. A significant relationship to annual evaporation can only be distinguished in the extratropical regions.

We further compare the same meteorological variables to the speleothem $\delta^{13}C_c$ data (Fig. 8, STab. 1). A significant relationship is only found with temperature in the extratropical region (Fig. 8c), with increasing temperatures corresponding to more depleted $\delta^{13}C_c$. $\delta^{13}C_c$ is also found to be enriched with altitude ($R^2 = 0.23$, $p < 0.02$, results not shown) in the extratropics. This $\delta^{13}C$-altitude relationship is not found in the other latitudinal bands.

The spatial testing shows globally strong relationships between $\delta^{18}O_{dweq}$ to environmental factors, in particular to altitude, temperature, precipitation, and evaporation. The spatial relationships between speleothem entity mean $\delta^{13}C_c$ and

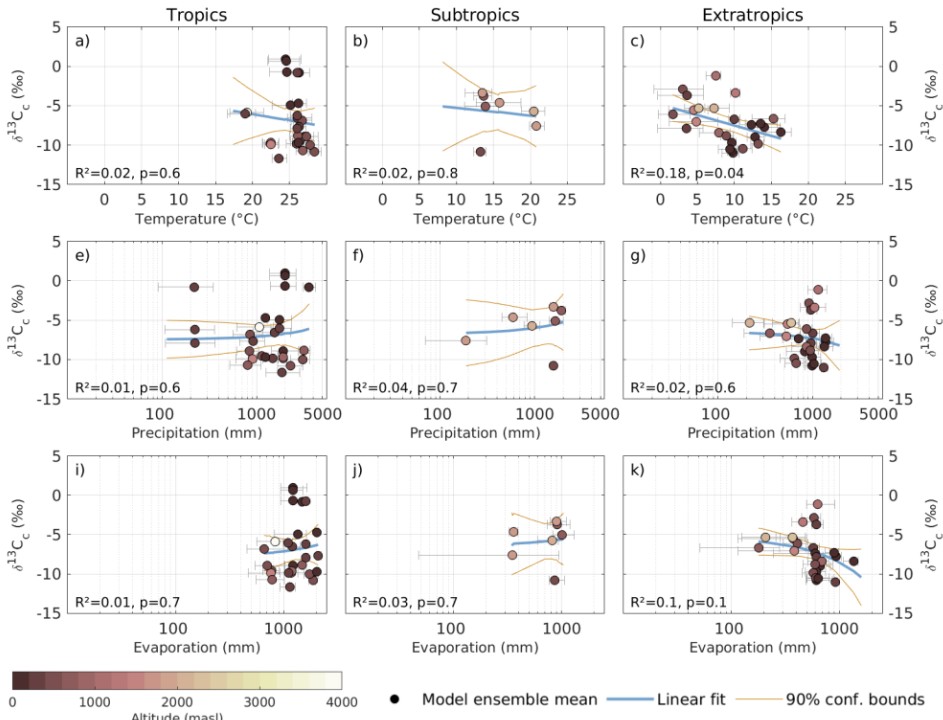

**Figure 8.** Climate-dependence of carbon isotope variability. Shown are simulated ensemble-mean temperature (a-c), precipitation (d-f) and evaporation (g-i) plotted against speleothem $\delta^{13}C_c$. In d)-i) altitude information is applied as shaded colors. We used linear regressions in all plots, however, these appear curved in a semi-logarithmic plot as used for precipitation and evaporation.

meteorological variables from model ensemble mean (Fig 8) only show clear relationships in the extratropical region, but not on a global scale.

### 4.3 Variability on different time scales

We compare the variance distribution in oxygen and carbon isotopes over all speleothems. This is a useful measure of how
climatic and environmental factors influence the proxies to a different extent. Additionally, different simulations have very different representations of variability across different timescales. This behaviour can be explored by calculating power spectral densities (PSD) of the simulated and recorded isotopes averaged globally.

Fig. 9a) provides the spectral ratio of the two isotopes after detrending the irregular time series. A flat spectral ratio at $\sim 1$ would indicate same levels of variability for both isotopes on all timescales. The spectral ratio here shows higher variability of
$\delta^{13}C_{speleo}$ on all timescales, however, for periods smaller than 10 years, the variability of both isotopes is more similar. This is supported by the total variance of the isotopes over the complete period. $\delta^{13}C_{speleo}$ shows a much higher total variance with a median of $0.46\%o^2$ (0.38, 0.6) compared to $\delta^{18}O_{speleo}$ with a median variance of $0.11\%o^2$ (0.08, 0.12).

Fig. 9b) shows the measured average PSD of $\delta^{18}O_{speleo}$ divided by the simulated average PSD at annual resolution, and Fig. 9c) by the average PSD of $\delta^{18}O_{sim}$ at record resolution. A spectral ratio larger than 1 indicates higher variability at
the timescale of the recorded $\delta^{18}O_{speleo}$, whereas spectral ratios smaller than 1 indicate higher variability of the simulated $\delta^{18}O_{sim}$. The spectral ratios between $\delta^{18}O_{speleo}$ and simulations at the cave locations at annual resolution show lower

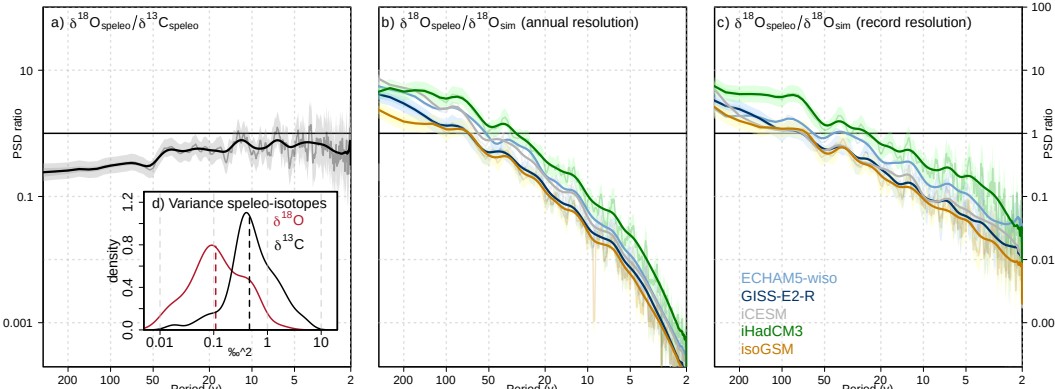

**Figure 9.** Spectral ratios of isotopes in speleothem and simulation on different timescales as shown by the ratios or mean power spectral densities (PSD): a) spectral ratio between speleothem isotopes ($\delta^{18}$O/$\delta^{13}$C). b-c) spectral ratio over all cave locations for $\delta^{18}$O$_{speleo}$ and $\delta^{18}$O$_{sim}$ per simulation (model-colors). In b) we show the spectral ratios of $\delta^{18}$O$_{speleo}$ to $\delta^{18}$O$_{sim}$ down-sampled to the individual records' resolution and in c) the simulated annually. The full spectra are shown in faded colors and a smoothed spectrum in black or the model colors. d) Variance of detrended $\delta^{18}$O$_{speleo}$ (red) and $\delta^{13}$C$_{speleo}$ (black) as measured in speleothem records. The dashed line indicated the median of the distribution.

variability in $\delta^{18}$O$_{speleo}$ compared to all models on decadal and shorter timescales although to different extents (Fig.9b). When considering the simulations down-sampled to record resolution, the result is similar but there is much lower variability in $\delta^{18}$O$_{speleo}$ at decadal and shorter timescales (Fig.9c). By down-sampling, the simulated spectra lose power in frequency on the decadal and shorter timescales, which is then reflected in higher spectral ratios. On decadal to centennial timescales,

however, $\delta^{18}$O$_{speleo}$ shows much higher variability than the modelled $\delta^{18}$O$_{sim}$, unaffected by the down-sampling.

Variability of $\delta^{18}$O$_{iw}$ is modelled differently in the simulations, as represented by the different levels of spectral power in the ratios. This difference is supported by the 5 area-weighted global variances of different magnitude, as well as the simulated variance in annual and down-sampled resolution as listed in Tab. 2. Small deviations between calculated variance and variance as area under the PSD can arise from the interpolation before the calculation of the spectra. The general order of

isoGSM having the highest and iHadCM3 the lowest power on shorter frequencies remains throughout, as can also be seen in the table with small deviations to the order. The global variance, however, is only partly represented at the cave locations. While isoGSM shows the highest variance globally and at cave locations, iCESM is of medium variance globally but has the smallest variance at cave locations. For unweighted isotopic composition, the order of simulations changes (results not shown).

The analysis suggests that variability in the simulated $\delta^{18}$O$_{iw}$ is represented differently in the simulations and that the order is not frequency-dependent. The recorded $\delta^{18}$O$_{speleo}$ shows more variability on centennial and less variability on decadal and smaller frequencies than the simulated, although to a different extent depending on the simulation.

### 4.4   Analysis of extreme events

To examine if there are factors that influence both $\delta^{18}$O$_{speleo}$ and $\delta^{13}$C$_{speleo}$ simultaneously, we analyze the similarity of

both signals. 86 % of the speleothems show significant correlation between both isotopes (results not shown). A different test for similarity is provided by checking for synchronous extreme events in the time series.





**Table 2.** Area weighted mean global $\delta^{18}O_{sim}$-variances as $var_{\text{global}}$ with 90% intervals of the distribution are given per simulation. Row two and three give the $\delta^{18}O_{sim}$-variance at the cave locations both in the annual resolution of the simulation, as well as down-sampled to the records resolution.

|  | ECHAM5-wiso | GISS-E2-R | iCESM | iHadCM3 | isoGSM |
|---|---|---|---|---|---|
| $var_{\text{global}}$ | 1.05 (0.15, 2.65) | 1.66 (0.19, 10.13) | 1.32 (0.07, 4.26) | 1.27 (0.13, 5.22) | 1.7 (0.16, 4.38) |
| $var_{\text{speleo}}$ (annual) | 0.66 (0.22, 2.31) | 0.66 (0.17, 3.98) | 0.66 (0.07, 3.1) | 0.37 (0.16, 1.83) | 1.36 (0.21, 9.83) |
| $var_{\text{speleo}}$ (down-sampled) | 0.19 (0.03, 1.53) | 0.25 (0.02, 1.88) | 0.15 (0.01, 2.05) | 0.15 (0.02, 1.42) | 0.29 (0.03, 8.25) |

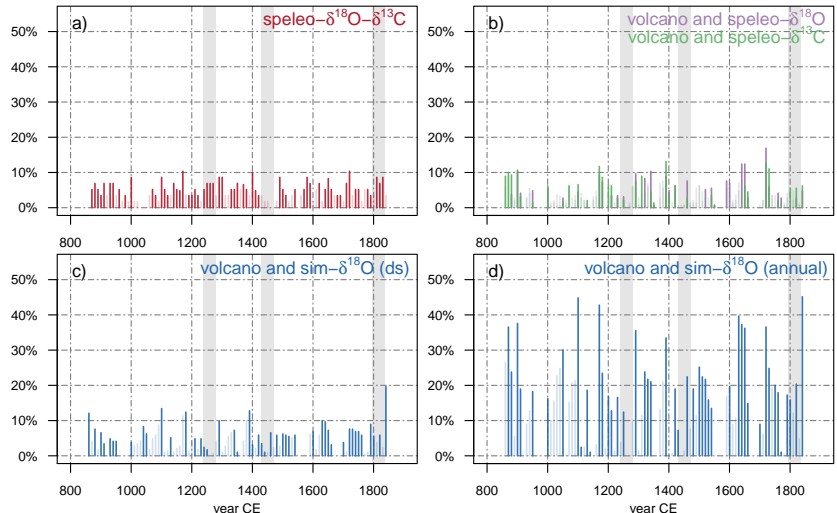

**Figure 10.** Synchronous events: a) the synchronous extreme events between $\delta^{18}O_{speleo}$ and $\delta^{13}C_{speleo}$ (red), b) the synchronous events between the speleothem isotopes (oxygen purple and carbon green), and volcanic eruptions as reconstructed by Crowley and Unterman (2013) or Gao et al. (2008) (depicted in Fig. 1d)), c) and d) the synchronous extreme events between simulated $\delta^{18}O$ values at the cave locations of all simulations in down-sampled to record resolution and annual resolution respectively. Where occurrence of synchronous extreme events is significant with $\alpha = 0.05$, the bars are shown in dark colors, non-significant in transparent colors. The four light grey bars in the background of each plot show areas of high volcanic activity.

Fig. 10 shows the temporal distribution of extreme events globally. In Fig. 10a) we test for synchronous extreme events between the two isotopes. Despite the high number of significantly correlated oxygen and carbon isotopes within one record, global patterns are not visible with a maximum of 10% of speleothems showing synchronous extreme events between $\delta^{18}O_{speleo}$ and $\delta^{13}C_{speleo}$ at the same time over the last millennium. Synchronous events are also not higher in time pe-

5 riods of strong volcanic eruptions (indicated by grey bars).

By analyzing synchronous events between volcanic eruptions as reconstructed by Crowley and Unterman (2013) and Gao et al. (2008) (Fig. 1d) and $\delta^{18}O_{speleo}$ as in Fig. 10b), up to 20% of speleothems exhibit extreme events at the same time as extreme volcanic eruptions. This share is higher than for the carbon isotopes, where up to 15% exhibit extreme events synchronous to volcanic eruptions. Both isotopes also show pronounced peaks occurring with different extreme volcano

10 events (indicated by grey background) but also for minor volcanic events.

To check if the speleothems used here can resolve global extreme events of short duration, we compare $\delta^{18}O_{sim}$ at the cave locations of each simulation to the volcanic forcing of the simulations (see Tab. 1). While up to 50% of $\delta^{18}O_{sim}$ at cave locations and annual resolution exhibit an extreme event at the same time as an extreme volcanic eruption (Fig. 10d), this



number largely decreases to less than half when the resolution of $\delta^{18}O_{sim}$ is decreased to that of the speleothems (10c). The number of pseudo-speleothems experiencing synchronous events to volcanic forcing (Fig. 10c) is more similar to that of the speleothems (Fig. 10b). Summarizing, we see no global temporal pattern of synchronous extreme events in both $\delta^{18}O_{speleo}$ and $\delta^{13}C_{speleo}$(Fig. 10a). A lower temporal resolution strongly decreases the ability of the modelled archive to resolve global

events like volcanic eruptions.

## 5   Discussion

### 5.1   Comparison of oxygen isotope variability in isotope-enabled models and speleothems during the last millennium

We found that the mean $\delta^{18}O_{sim}$ fields show global differences of $2.12\,‰$ between the models, that could mostly be attributed

to the global mean temperature differences $1.8\,K$ between the models. Similarly, most of the strong regional differences in $\delta^{18}O_{sim}$ between models could be explained by regional differences in simulated temperature (SFig. 3), as temperature was shown to be a major driver of $\delta^{18}O_{sim}$ (Fig. 5a). Specifically, we found less depletion of $\delta^{18}O_{sim}$ in isoGSM over Antarctica as an artifact of its numerical scheme used for moisture transport, and linked to extremely dry regions (Yoshimura et al., 2008). The warm bias in high latitudes for ECHAM5-wiso results is an underestimation of isotope depletion (Werner

et al., 2011, 2016). Overestimation of fractionation processes in iCESM during re-evaporation processes resulted in generally stronger depletion in $\delta^{18}O_{sim}$ (Brady et al., 2019). The cool bias in northern mid-latitudes in the GISS-E2-R model as found in Schmidt et al. (2014) resulted in more depleted $\delta^{18}O_{sim}$ in this region. Colder temperatures over Antarctica in iHadCM3 explained partly why isotopic signatures are a lot more depleted than in the other simulations. Compared to historical ice core data, iHadCM3 mean isotopic signatures above Antarctica indicated realistic values (Tindall et al., 2009) suggesting

that the colder Antarctic conditions modelled by iHadCM3 may be more consistent with reality than the multi-model mean. Even though this study mostly focused on a terrestrial mid-to-low latitude archive, local differences of modelled isotopic signatures in the Antarctic may have an influence on isotopic representation in the general circulation of the models.

At the cave locations, the spread between the simulations yielded $4.51\,‰$ (90% CI: 3.96, 4.79) in median (Fig. 3), while the median offset between simulated and speleothem $\delta^{18}O_{dweq}$ was around $-0.38\,‰\,(-0.8, -0.23)$(Fig. 4. This means that

even though simulations differed strongly in some regions, using multiple models can be sufficient to average out the offset at cave locations. The offsets to the speleothems were in agreement with those found by Bühler et al. (2021) who compared the SISALv2 database to the iHadCM3 last millennium simulation. Median offsets to the speleothem records were small for all models, where differences spatially around the globe (Fig. 3) reflected both the internal variability of models, as well as global differences between the models. For example, ECHAM5-wiso showed the highest values for $\delta^{18}O_{sim}$ in the global

mean and also a more positive offset distribution than the other models.

Our analysis suggests that a multi-model approach is advisable whenever comparing mean modelled values to data. Even though global mean $\delta^{18}O_{sim}$ values may be comparable, local and regional temperature estimates, and, therefore, modelled $\delta^{18}O_{sim}$ values can vary strongly and deviate between models. Even though isoGSM displayed the lowest offsets between $\delta^{18}O_{sim}$ and $\delta^{18}O_{speleo}$ (Fig. 4), additional processes between meteoric water above the cave and drip water may again

influence this mean offset. Still, the multi-model offset comparison justified the use of the multi-model mean at cave locations in the following spatial analysis.

We found significant relationships between all considered climatic variables (temperature, precipitation, and evaporation), and simulated $\delta^{18}O_{sim}$ and mean $\delta^{18}O_{dweq}$ (Fig. 7). These relationships were more distinct over latitude bands, which is in line with the effects described by Dansgaard (1964). However, the relationships between speleothem mean $\delta^{13}C_c$ and

meteorological variables from model ensemble mean (Fig 8) were less clear.



Global studies that have evaluated $\delta^{18}O_{speleo}$ isotopic signatures using climate models already exist (Bühler et al., 2021; Comas-Bru et al., 2019; Midhun et al., 2021), where regions with shared climatic features showed stronger relationships. For example, Baker et al. (2019) focused on specific climate zones by comparing drip-water measurements to precipitation $\delta^{18}O$ measurements and was able to identify temperature zones for which mean measured $\delta^{18}O$ or $\delta^{18}O_{iw}$ was most similar to

$\delta^{18}O_{speleo}$. In our study, we found stronger relationships to climate variables in the latitude bands than compared to a global assessment as in Bühler et al. (2021). Analyzing regions with high data density and similar climate patterns, we found even stronger temperature relationships (SFig 7 and SFig. 8). Still, local particularities, such as large elevation difference over short distances, could not be resolved properly by the simulations and explained many of the very strong outliers, especially in the tropics.

Comparing a last century subset of the SISALv1 database by Fohlmeister et al. (2020) with our last millennium SISALv2-subset yielded similar results for precipitation and altitude relationships to $\delta^{13}C_{speleo}$ (SFig. 6). In contrast to our latitudinal approach, they compared their $\delta^{13}C_{speleo}$ dataset to temperature on a global scale. However, they neglected clusters of speleothems, where known carbon isotope governing factors other than temperature play an important role for the carbon isotope composition (e.g. high amounts of precipitation, known cave specific particularities and processes, or temperatures

close to the natural limit of vegetation ($< 5°C$)). The remaining records with temperatures between $\sim 7$ to $27°C$ showed a positive trend between $\delta^{13}C_{speleo}$ and temperature. This trend is in contrast to our observation based on clustering the records according to latitudinal bands. With this approach, we find no relationship between $\delta^{13}C_{speleo}$ and temperature for the tropics and subtropics, but a clear inverse relationship is observed for the extratropical records.

Higher cave site elevation coincided significantly with more depleted $\delta^{18}O_{dweq}$. For $\delta^{13}C_{speleo}$, local studies exist that

predict an increase in $\delta^{13}C_{speleo}$ with higher altitude (Johnston et al., 2013). However, for the global last millennium subset of the SISALv2 database more entities with carbon measurements in higher altitudes are needed to see a potential global relationship, in addition to the relationship we find in the extratropical latitude bands.

## 5.2  Can models reproduce variability archived in speleothems?

For all simulations, temperature variability was the dominant driver in $\delta^{18}O_{sim}$ at high latitudes and precipitation variabil-

ity at low latitudes (Fig. 5). At the cave sites, model-internal regional variability as well as the records' age uncertainties substantially decreased correlation estimates. We observed that the sign of the correlation estimated between simulated temperature and $\delta^{18}O_{sim}$ agreed at 60% of cave locations for 3 and more simulations, and at 26% for 4 and more simulations. For correlation estimates to precipitation, this was true only at 11% or 1% of locations, respectively. When compared to measured $\delta^{18}O_{speleo}$ , we found more significant temporal correlation estimates to modelled temperature than to modelled

precipitation. This could in part be explained by global temperature responses to e.g. volcanic forcing being more uniform between model ensemble runs compared to precipitation responses, which depend strongly on regional particularities. Regions with high inter-model climate variable spread (as can be seen in SFig. 2) also coincided with regions of the least significant correlation estimates to simulated temperature and the least agreement between the climatic drivers of $\delta^{18}O_{sim}$ .

When looking at variability specifically at the cave site locations, we saw that for all speleothems where both $\delta^{18}O_{speleo}$

and $\delta^{13}C_{speleo}$ are available, $\delta^{13}C_{speleo}$ appeared to be more variable on average on all timescales (Fig 9d) with an increasingly higher variability compared to $\delta^{18}O_{speleo}$ towards centennial timescales and longer (Fig 9a). Within 86% of all speleothems, where $\delta^{18}O_{speleo}$ and $\delta^{13}C_{speleo}$ are provided, carbon and oxygen showed significant correlations. Jointly explained variance in the isotopic signal could point to common climatic drivers. However, the amplified variance on long timescales in the carbon isotope ratio could not only hint at changes in the water cycle but also land surface processes such

as soil formation or vegetation changes. Considering more terrestrial archives as well as trace elements stored within the speleothem may help to better disentangle the climatic and environmental signals. On decadal and shorter timescales, where





the meteoric and seasonal vegetation isotopic signal is mostly smoothed by the karst system, higher variability in $\delta^{13}\mathrm{C}_{speleo}$ may result from the stronger isotopic fractionation for carbon compared to oxygen in precipitated calcite (Polag et al., 2010; Hansen et al., 2019).

Climate models reflected $\delta^{18}\mathrm{O}_{sim}$ variability at cave locations to different extents. A clear offset between the models could
be found on all timescales. We found no relationship between spatial resolution of the model and the variability of isotopic composition of precipitation. The higher-resolution run of iCESM and the lower-resolution run of iHadCM3 seem to show similar variability globally and the lower-resolution ECHAM5-wiso shows even higher variability at cave locations. Due to the strong impact of temperature and precipitation on $\delta^{18}\mathrm{O}_{sim}$ variability we expect that this difference in isotopic variability also stems from the difference in the simulated climate. Further assessments using multivariate statistics are needed to firmly
attribute the impact of climate on recorded isotopic variability.

The lower temporal resolution of speleothem records largely explained the model-data mismatch on decadal and shorter timescales. Slow growth rates and limited sampling resolution lead to averaging effects which then lead to lower variability on shorter timescales. Simple karst-filters of a realistic transit time of $\sim 2.5$ yr as in SFig. 9b) (as used in Bühler et al., 2021; Midhun et al., 2021; Dee et al., 2015) showed that variations in models and speleothems on these shorter timescales are very
similar, if accounted for. Expert knowledge of the local cave hydrology is, however, needed for a more detailed assessment on which model reflects $\delta^{18}\mathrm{O}_{sim}$ variability best on decadal and shorter time scales compared to speleothems and may still be restricted by karst and cave internal processes that effectively limit the sampling of climatic signals.

On decadal and longer timescales models seemed to underestimate $\delta^{18}\mathrm{O}_{speleo}$ variability, although to different extents, with isoGSM showing the highest variability and the smallest model-data mismatch on decadal and longer timescales. The
model-data mismatch that we observed between speleothems and $\delta^{18}\mathrm{O}_{sim}$ starting at decadal timescales is in agreement with previous studies as by Laepple and Huybers (2014b), however, it is worth mentioning that speleothems may also be capable of enhancing climate-driven changes of $\delta^{18}\mathrm{O}$ and $\delta^{13}\mathrm{C}$ by cave-specific processes, resulting in higher variability on decadal and longer timescales. However, this has to be verified by future studies. Under the assumption that variability on decadal and longer timescales is recorded correctly by speleothems, the iCESM model showed the strongest variability mismatch and
isoGSM the smallest. A model-data match on decadal and shorter timescale depends strongly on cave-hydrology processes that additionally dampen the meteorological signal.

Cave locations were not necessarily reflective of mean annual $\delta^{18}\mathrm{O}$ variability globally - at least not in the model simulations. Simulations that showed generally high variability at cave locations at speleothem resolution also tended to be more variable globally and vice versa for low variability. However, simulations that were less variable at cave locations than others
can still be more variable globally. In our case, this trend could likely be attributed to the bias of geographic locations of the cave sites as the models mostly show high variance in $\delta^{18}\mathrm{O}_{sim}$ in very dry regions and around the regions of the inter-tropical convergence zone.

### 5.3  Can external forcings be resolved by speleothems?

We found that 86% of speleothems have a significant temporal correlation between speleothem oxygen and carbon isotopes,
with 47% even showing strong significant (anti-) correlations of $|c| > 0.5$. The co-variability of both isotopes has been studied for a very arid region stalagmite by Fohlmeister et al. (2017) who also found strong correlation between both isotopes. High correlation between the isotopes could hint at kinetic isotope fractionation effects (Hendy, 1971). Fohlmeister et al. (2017) attribute increased correlation to times of strong variations in cave-internal processes triggered by variations of external conditions. This simultaneity agrees with our findings that generally no extreme event in isotopes precedes the other, which
can, however, also be attributed to low sampling resolution.





Changes in temperature or precipitation due to aerosol forced cooling have been analyzed in a $\delta^{13}$C record as signs of volcanic signatures of speleothems (Ridley et al., 2015). Growth rate changes (Baker et al., 1995) or the measurement of trace elements such as sulphur (Frisia et al., 2008) are other techniques to detect volcanic signals in speleothems but they generally require up to sub-annual resolution records. In our global analysis of 58 $\delta^{13}$C$_{speleo}$ and 89 $\delta^{18}$O$_{speleo}$ records, we

saw no significant increase in extreme events in the isotope records coinciding with major volcanic eruptions. The individual isotopes yielded more distinct signatures of volcanic eruptions with up to 20% (15%) of speleothems recording synchronous extreme events in $\delta^{18}$O$_{speleo}$ ($\delta^{13}$C$_{speleo}$) and a volcanic eruption, very similar to $\delta^{18}$O$_{sim}$ at record resolution. Both stayed, however, well below the possible simulated detection at cave locations under annual resolution of up to 50%. The comparison to the synchronous events to the down-sampled $\delta^{18}$O$_{sim}$ showed that the ability to capture events such as volcanic eruptions

strongly decreases with record resolution. The attribution of specific peaks in speleothem data to volcanic events needs caution because of age-uncertainties and other possible explanations for the changes e.g. human settlements close-by (Baker et al., 1995). Increasing the bin size to the average age-uncertainty within the last millennium sub-set of the SISALv2 yielded the same results (SFig. 11).

Solar variation is another external forcing which is often invoked as an influence on the monsoon cycle that has been

investigated using speleothem records (Neff et al., 2001; Lone et al., 2014; Cosford et al., 2008). In these studies, it is also standard to use high-resolution speleothems with a lowest resolution of about 5 yr for Lone et al. (2014). We repeated the analysis for Fig. 10 to analyze synchronous extreme events of $\delta^{18}$O$_{speleo}$ and $\delta^{13}$C$_{speleo}$ to the total solar irradiance input (SFig. 10). While the general effects of decreased detectability with decreased resolution are also visible, the overall detection of extreme solar irradiance was much weaker than for volcanic eruptions. This was not only true at cave locations

but globally when comparing simulated surface climate variables to solar variations (results not shown). As solar variation on this timescale is mostly cyclical compared to random extreme volcanic eruptions (compare Fig. 1), these results were to be expected from the methods used and are not in contradiction to the literature.

Summarizing, the comparison with the $\delta^{18}$O$_{sim}$ showed that cave locations are in general suitable to detect climatic changes due to changes in volcanic or solar forcing. Even though speleothems are highly resolved archives with little age-

uncertainties compared to other archives, the median resolution during the last millennium of 6.08 yr (4.07, 7.85) was not enough to resolve changes in $\delta^{13}$C$_{speleo}$ or $\delta^{18}$O$_{speleo}$ due to potential solar or volcanically–induced climatic changes. Karst-mixing effects which further dampen the signal, as discussed for the variability on shorter timescales for SFig. 9, may decrease the ability to detect these changes, if they exist, even further.

## 5.4   Limitations

A current weakness of this type of analysis is that we only compared simulated oxygen isotopes of PMIP2/PMIP3 generation models to archived oxygen isotope signals. For carbon isotopes no simulated values are available yet. For iCESM, a carbon cycle in the ocean exists (Jahn et al., 2015) and great effort is put into the incorporation of a carbon-cycle to iGCMs by the scientific community. The multi-model comparison in this study only included how different models represent isotopes. However, parameter and tuning choices within one model, especially in the cloud and convection scheme, have a strong

imprint on $\delta^{18}$O signatures (for example Nusbaumer et al. (2017) for iCESM, Field et al. (2014) for GISS-E2-R). To further systematically explore and constrain modelled $\delta^{18}$O, a multi-model ensemble under different model set-ups will be needed.

We also only looked at carbon and oxygen isotopes as possible proxies for climatic changes. In contrast to e.g. ice-cores, speleothems do not directly record precipitation $\delta^{18}$O but instead archive $\delta^{18}$O with additional fractionation processes. Both $\delta^{18}$O and $\delta^{13}$C undergo fractionation processes which can be influenced by various cave-internal processes (Lachniet, 2009;

Fairchild and Baker, 2012; Hartmann and Baker, 2017). Here, the isotopes in the drip-water are influenced by many fraction-ation processes that are not climate-related (Dreybrodt and Scholz, 2011; Fohlmeister et al., 2020). Considering additional





processes such as prior calcite precipitation (PCP) or other geochemical climate-related proxies can help to decipher the climatic signal from karst- or cave-internal processes (Kaufmann, 2003; Schwarcz et al., 1976; Owen et al., 2016; Tremaine and Froelich, 2013; Noronha et al., 2014). Especially sulphur proved to be a valuable tracer in detecting volcanic eruptions (Frisia et al., 2008). This possible offset between model and data is, however, assumed to be the same for all simulations and

does not affect the multi-model analysis.

When comparing the speleothem isotopes to volcanic data, we note that there are more recent volcanic reconstructions available, which suggest a modification to the timing or magnitude of last millennium eruptions (Sigl et al., 2015). Given the temporal resolution of the speleothems, changes in timing of volcanic events would impact the comparisons of model data to only those speleothems with very high temporal resolution. As only the timing of the extreme event is compared, a change in

magnitude would be irrelevant. As for the comparison with the simulated data, only the response of the model to any given eruption is important (Colose et al., 2016b), as we did not compare the timing of extreme events in speleothems to those in the simulation.

Compared to the capability of speleothems to cover complete glacial-interglacial cycles, we only looked at very short timescales when focusing on the last millennium and what drives variability on decadal to centennial timescales instead. The

last millennium is considered a relatively stable time period and climatic changes might not be strong enough to be fully captured by speleothems. On longer timescales, speleothems may still be a good archive to capture these larger changes (Genty et al., 2006).

## 6  Conclusion

We presented a multi-model comparison over five last millennium isotope-enabled simulations (ECHAM5-wiso, GISS-E2-

R, iCESM, iHadCM3 and isoGSM) and compared their representation of isotopic signatures in mean and variability to paleoclimate data from a large speleothem database (a last millennium subset of SISALv2). We found that $\delta^{18}O_{sim}$ differed substantially between models on a regional scale as well as at speleothem cave sites. This could mostly be attributed to differences in modelled temperature between models. This effect can be compensated by using the multi-model mean. The isoGSM simulation showed the lowest absolute mean offset to the speleothems at cave locations, while all other simulations

show only slightly higher offsets. Variability on decadal and longer timescales in the speleothems was higher than indicated by models and was also best represented by isoGSM, which, however, still underestimated variability on these timescales. No relationship was found between the spatial resolution of the models and their variability of the isotopic composition of precipitation. In all models, temperature was driving $\delta^{18}O_{sim}$ variability in high latitudes and precipitation in low latitudes. At cave site locations in particular, which are mostly located in low- to mid-latitudes, models agreed more on temperature

being the driving factor of water isotope variability than on precipitation.

Dividing the global set into latitude bands, we were able to distinguish temperature and altitude relationships for both the oxygen and the carbon isotopes, as well as significant relationships for $\delta^{18}O_{dweq}$ to other simulated climate variables. While most records showed significant correlation between the two isotopes, using both isotopes to gain more information than just from one remained difficult. Especially, variations in solar and volcanic forcing are not imprinted in either the single isotope

or the pair on a global scale. Many archive limitations could, however, be attributed to the low resolution of the data-set compared to the processes expected to be resolved.

Our analysis encourages the use of multi-model means whenever possible as already suggested by other studies (Colose et al., 2016a). From the point of model evaluation, the incorporation of different archives with higher resolution (e.g. corals, trees, ice cores as in the iso2k database (Konecky et al., 2020)) and with the help of improved proxy system models may

provide further insight into why offsets between models can be so large regionally. From a speleothem perspective, within-





cave and between-cave variability comparisons using both isotopes will help to understand the recorded signal better and give higher confidence.

Future multi-model comparisons of isotope-enabled models for other time periods are required to further evaluate biases in models, as well as comparisons to $\delta^{18}$O-archives of all kinds, as to help to constrain models. If carbon isotopes are included in a multitude of water-isotope-enabled models as well, this needs to be repeated for both isotopes, to gain a deeper understanding of the underlying concept in what influences variability and co-variability of isotopes in speleothems.

**Code and Data availability.**

Code to reproduce figures and analyses will be made available in the final manuscript on GitHub. Model data as csv-files with output at cave locations (ECHAM5-wiso, GISS-E2-R, iCESM, iHadCM3, isoGSM) at annual and record resolution, as well as monthly fields of surface temperature, precipitation, isotopic composition of precipitation and evaporation or latent heat respectively for all simulations will be made freely available on Pangaea in the final manuscript. The SISAL (Speleothem Isotopes Synthesis and AnaLysis Working Group) database version 2 (SISALv2) is publicly available through the University of Reading repository at http://dx.doi.org/10.17864/1947.256 (Comas-Bru et al., 2020a). We use R for the data analysis (R Core Team, 2020a). The main packages are tidyverse (Wickham et al., 2019), ncdf4 (Pierce, 2019), ggplot2 (Wickham, 2016), raster (Hijmans, 2020), zoo, (Zeileis and Grothendieck, 2005), plyr (Wickham, 2011), and (Wickham et al., 2021). We use the nest R package (https://github.com/krehfeld/nest Rehfeld et al., 2011; Rehfeld and Kurths, 2014) and the PaleoSpec package (https://github.com/EarthSystemDiagnostics/PaleoSpec Kunz et al., 2020).

**Author contributions.** JB, JA, FL and KR designed this study. JB and JA wrote the paper and created the figures. FL and JF contributed with data interpretations. KR, JS, KY, MM, ALG and MW contributed with model data that JB and JA collected and standardized. JB prepared the speleothem data. JB, JA, KR, FS, JF, JS, KY, ALG, MM and MW contributed to the revisions of the manuscript. All authors approved of the final version of the paper.

**Competing interests.** The authors declare that they have no conflict of interest.

**Acknowledgements.** This study includes data compiled by SISAL (Speleothem Isotopes Synthesis and Analysis), a working group of the Past Global Changes (PAGES) project. PAGES received support from the Swiss Academy of Sciences and the Chinese Academy of Sciences. The project is in part inspired by discussions at the SISAL 4th workshop: Exploiting the SISALv2 database for evaluating climate processes, Xi'an, China, 14-18 October 2019. Nils Weitzel, Elisa Ziegler, Beatrice Ellerhoff, Qiong Zhang, Nikita Kaushal, Natasha Sekhon, Valdir Felipe Novello, Jon Baker, Ny Riavo Voarintsoa and Yuval Burstyn for helpful advice, comments and discussion on text and figures. Parts of the computations were enabled by resources provided by the Swedish National Infrastructure for Computing (SNIC) at the National Supercomputer Centre (NSC), partially funded by the Swedish Research Council through grant agreement no. 2018-05973.

**Financial support.** We acknowledge financial support from the Swedish Research Council (Vetenskapsrådet; grant nos. 2013-06476 and 2017-04232), the Deutsche Forschungsgemeinschaft (grant nos. 316076679 and 395588486), the Bun-





desministerium für Bildung und Forschung through the PalMod project (grant no. 01LP1926C), and the Swiss National Science Foundation (SNSF; grant no. P4P4P2_186693).

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
