# Peer review of "Investigating stable oxygen and carbon isotopic variability in speleothem records over the last millennium using multiple isotope-enabled climate models"

_Climate of the Past, 2021_

## Author Response (AR1)

**Final response to the reviewers' comments: Investigating oxygen and carbon isotopic relationships in speleothem records over the last millennium using multiple isotope-enabled climate models (cp-2021-152)**

Janica C. Bühler, Josefine Axelsson, Franziska A. Lechleitner, Jens Fohlmeister, Allegra N. LeGrande, Madhavan Midhun, Jesper Sjolte, Martin Werner, Kei Yoshimura, and Kira Rehfeld

April 19, 2022

**Summary of changes**

Dear Qiuzhen Yin,

In response to the suggestions by the two reviewers, we implemented the following changes to the manuscript:

- change the title to **"Investigating stable oxygen and carbon isotopic variability in speleothem records over the last millennium using multiple isotope-enabled climate models"**

- carefully restructure and rewrite the introduction to better motivate our research and change the conclusion alongside

- revise the method section to motivate and highlight the influence of different weighting procedures to the results more thoroughly,

- clarify the use of the word "offset" between different simulations or between simulation and record where necessary,

- revise the data section in conjunction with the discussions to clarify the differences between model simulation setups and boundary conditions,

- revised Fig. 1b to include the drip-water converted speleothem $\delta^{18}O$,

- change Fig. 5 to include Fig. 5c, which will strengthen our discussion on major climatic drivers,

- revise the discussion to include fundamental isotopic effects, different climatic backgrounds,

- revise the text throughout the manuscript to clarify statements,

- fix formatting where necessary.

A detailed response to the helpful remarks of the referee is given below.

**1 Reply to the first reviewer**

(Original report cited in italics)

*1) **Infiltration adjusted precipitation weighting:** d18Oiw is an interesting method and I think it be beneficial to discuss it a bit more thoroughly. 1) A stronger justification for its use in this paper would be useful, such as more clearly stating why the results are more realistic for comparing to speleothem data. 2) A more detailed description of this method regarding how it differs from d18Op would be useful (highlighting the strong role played by evaporation). 3) Is this method justifiable over marine environments? I understand that it is preferable for understanding infiltrating water into a cave system, but I wonder if it artificially elevates the importance of evaporation over marine environments where there is always available water to evaporate? Since a key finding in this paper is that temperature drives speleothem values even at lower latitudes, I wonder if this takeaway is at least somewhat attributable to an artificially heightened dependence on temperature (via evaporation) at lower latitudes?*

Thank you for this interesting comment especially with regard to evaporation.

1+2) We will add a stronger justification in the methods section as to why we use $\delta^{18}O_{iw}$ instead of annual-mean $\delta^{18}O_{sim}$ and we will highlight the role of evaporation in the method section as follows:

"... $\delta^{18}O_{speleo}$ **forms from drip water that reaches the cave, which is the precipitation water minus all water that evaporates. When comparing modelled to speleothem isotopes it is more realistic to weight the modelled $\delta^{18}O_{sim}$ by** precipitation minus evaporation amount (infiltration adjusted precipitation weighting, $iw$) to obtain annual values. **Simply using the annual mean $\delta^{18}O_{sim}$ would overemphasize the isotopic composition of seasons where little to no precipitated water reaches the cave as drip water due to strong evaporation above the cave. The weighting therefore automatically focuses on the local seasonal composition of SWI in precipitation that will theoretically reach the cave and form a speleothem**. [...] As isotopic fractionation **also** occurs during evaporation from the soil, models where $\delta^{18}O_{sim}$ is also available for soil layers, would be more realistic to compare to speleothem data. However, these were only available for a few simulations. Using infiltration-weighted $\delta^{18}O_{sim}$, therefore, offered a more equal handling of the data **while maintaining the large ensemble** and enabled a better comparison of results. ..."

3) In this study, we use $\delta^{18}O_{sim}$ in precipitation. When studying marine environments infiltration weighting of $\delta^{18}O_{sim}$ is not the right variable to look at, but instead one should focus on the $\delta^{18}O$ in seawater. Nonetheless, your thoughts on the artificially highlighted dependence on temperature through evaporation are justified. To this end, we added figure A1. The figure shows that the infiltration weighting of the SWI artificially lowers the dependence of $\delta^{18}O$ on temperature, as it puts a weight on months with high precipitation and little evaporation instead of months with high evaporation. The correlation estimates are smaller globally for infiltration weighted data. When looking at correlation maps for precipitation (not shown) the correlation estimates are increased through infiltration weighting in regions where high precipitation falls in months with lower temperature, and decreased in regions where high precipitation falls in months with high temperature.

Action: Done.

[Figure]

Figure A1: Correlation map between simulated $\delta^{18}O$ (a) or $\delta^{18}O_{iw}$ (b) to temperature. c) shows the difference between the two correlation maps.

*2)* **"Offset"**: *Throughout the paper, the term "offset" is used, but is generally loosely defined. It will help the readers to be explicit in the definition of this word. I was confused at times and wondered if this term referred to a) the difference of an individual model's values from the multi-model mean or b) the difference between model values (either individual or multi-model means) and speleothem values.*

Thank you for pointing this out. In the manuscript, we have used the term "offset" in both circumstances a) and b) mentioned by the reviewer. However, we agree with the reviewer that this can cause confusion and hence decrease readability. When revising the manuscript, we plan to keep the term "offset" when referring to the differences of individual models to multi-model mean, and use the terms "differ" and "deviate" when referring to the difference between model and speleothem values.

Action: Done.

*3)* **Temporal and spatial averaging in the models**: *Please include more discus-*

*sion on the uncertainty related to your choices regarding model averaging at speleothem locations. Annual mean model results are taken from a single gridbox that most closely corresponds spatially with the speleothem record – did I interpret this averaging method correctly? This paper will be strengthened if it includes some more discussion on the ways in which the choices in averaging impact the results – 1) How might the results change if instead of annual averages, seasonal averages (i.e., wet season, summer season, etc.) are used? Or if instead of a single gridbox, a larger spatial averaging region (i.e., also including all adjacent gridboxes) was used?*

Thank you for your interesting thoughts. 1) In this study, simulation data was available at monthly resolution. This allowed us to do infiltration weighting on the time series and calculate an annual value that emphasizes the season with the highest amount of precipitation that is not evaporated. In a global analysis, this results in different months dominating the annual isotopic value at each gridbox depending on local climate conditions. The same however is achieved, when taking the annual mean of monthly mean $\delta^{18}O_{sim}$. This averaging would over-represent specific months with only little precipitation. As all averaging processes include such seasonal biases, we chose the weighting since this theoretically correspond best to cave systems.

2) The averaging method is not choosing the gridbox that most closely corresponds spatially, but we extract simulated values by bi-linear interpolation, already taking into account neighbouring gridboxes. This is described in Sec. 3.2 Data processing on page 10 line 25. However, other extracting methods such as kriging interpolation have already been tested in other studies (Latombe et al., 2018). They show that bilinear or bicubic interpolation techniques distort either the temporal variability or the values of the response variables. We will add a short paragraph of the impact of our interpolation method to our results in the limitations-section.

Action: We added the following sentence to the methods section:
Action: "... We note that this bi-linear interpolation can, however, influence the temporal variability or the values of the response variables (Latombe et al., 2018)..."

**Reply to the second reviewer**

*1) The subject of this manuscript is unclear. The current models cannot simulate the carbon isotope, how to investigate the carbon isotope using models? Thus, the title is inappropriate. This work cannot explain the relationship between the oxygen and carbon isotopes. Another option focus on the ensemble mean of the multiple GCMs. The highlights is likely derived from the differences and commonalities between the ensemble mean and each member.*

We thank the reviewer for this comment, which can help us clarify the manuscript. Indeed, simulated carbon isotopes are not implemented in the models. However, we

can compare simulated climatic variables, such as temperature, precipitation and evaporation, to the speleothem data, as these variables have shown to partially control speleothem $\delta^{13}C$ (Fohlmeister et al., 2020; Novello et al., 2021). Thus, our aim is not to investigate the direct relationship between speleothem $\delta^{18}O$ and $\delta^{13}C$, but rather their climatic controls and their response to forced events, such as volcanic eruptions and changes in solar forcing. Following the comment raised by the reviewer we will modify the title to **"Investigating stable oxygen and carbon isotopic variability in speleothem records over the last millennium using multiple isotope-enabled climate models"**. We intend to make sure that the introduction clearly states what relationships we are investigating and how this is performed in regard to the model simulations.

Action: Done.

*2) The mechanism and the reason need to be further explored. The advantage of climate model is to explore the mechanism. How does the temperature affect the oxygen isotope signature? What's the feedback? How does precipitation amount impact the water isotope at low latitudes?*

We agree with the reviewer that model simulations are a great tool for exploring and understanding mechanisms in the climate system. From the implementations of stable water isotopes in each of the individual models, their individual performance and potential biases in $\delta^{18}O_{sim}$ compared to observations and/or proxy data is already established (e.g. Bühler et al., 2021; Comas-Bru et al., 2019; Midhun et al., 2021). Following the comment raised by the reviewer, we intend to be more explicit in our explanations of the isotopic signatures and the mechanisms behind them theoretically, based on our findings. The fundamental processes causing isotopic fractionation effects by changes in temperature, precipitation amount, geographical location, circulation patterns and seasonal effects are well-established in previous literature (Dansgaard, 1964; Rozanski et al., 1992). Here, we do not aim to explain the mechanics and dynamics in all five simulations in relation to the speleothems. We rather find and investigate where simulations and proxy data match – or don't, and whether simulations yield consistent patterns themselves.

However, also following the suggestions of reviewer 1, we will include more discussion throughout the manuscript on these fundamental isotopic effects as described by Dansgaard (1964) and Rozanski et al. (1992), and elaborate more on where we see the effects (like amount and continental effect) in proxy and model.

Action: We refer to isotopic effects as described by Dansgaard (1964) and Rozanski et al. (1992) more spread out through the discussion.

*3) The details are needed be carefully checked and the logic and legibility should be further improved.g. It is too arbitrary to obtain the conclusion of the "major driver" from a correlation map in the climate model study. If the differences between the models*

*is so large, how to definite that the ensemble mean is climate signal or noise.*

We agree with the reviewer, that more model diagnostics need to be checked in order to obtain a coherent picture of which variables drive $\delta^{18}O$ in the model world. However as for this analysis, only few variables were available for all models and even for evaporation, latent heat had to be used as a surrogate for some models. We will clarify in the discussion, that more variables need to be tested.

We also agree, that Fig. 5 is not sufficient to conclude on the major drivers. We revised Fig. 5 and added a Fig. 5c as in Fig. A6, from which conclusions can be drawn more easily. Before we were relying on the supplement SFig. 5, which show dominant regions for the variables temperature and precipitation. Additionally, we emphasise that Fig. 5 does not result from the correlation of the ensemble mean fields, but instead shows the mean of the correlation fields for each simulation. From SFig.5 as well as our agreement markers in Fig. 5 we show, that we do see a modelled climate signal.

We will carefully revise the sections, where we explain the correlation maps and better discuss our conclusions. Additionally we will highlight the need to analyse more variables in the discussion more clearly. A revised Fig. 5 is provided and explained in the Detailed Comments.

Action: We added Fig.5c to the manuscript and revised the sections where we discuss the figure

**2    Detailed Comments**

We sorted the detailed comments by their reference to lines in the manuscript. The reviewers are indicated as **R1** and **R2** respectively. Where both reviewers commented on the same section, we combined the answers such that both comments are addressed in one segment.

**R2** *The introduction is not focused. If possible, please highlight the importance of comparing simulated water isotopes with measured speleothem isotopes, illustrating the reasons for analysis from spatial, temporal and extreme events aspects.*

We will revise the whole introduction and highlight the innovation of our study more clearly. We will especially follow the advise from the reviewer to highlight (1) the importance of comparing simulated water isotopes with measured speleothem isotopes, (2) illustrating the reasons for analysis from spatial, temporal and extreme events aspects. Besides restructuring, we will include the following thoughts into the introduction:

1. Following the recommendations of PAGESHydro2k-Consortium (2017), proxy and model comparison should take place on equal ground. If we want to analyze the representation of the modelled hydrological cycle, archives of $\delta^{18}O$ are the most common. Comparisons need to take place on the $\delta^{18}O$ level, to avoid uncertainty through proxy calibration to specific desired variables and subjective interpretation.

2. Spatial and temporal consistency between modelled and archived data is to be critically evaluated (PAGESHydro2k-Consortium, 2017) unless externally forced e.g. through volcanic eruptions. Spatial and temporal inconsistencies can arise from model-topography or internal variability. Nonetheless, modelled temporal variability in the frequency domain can be evaluated using proxy data. Also global spatial patterns in models can be evaluated.

Action: We carefully revised the introduction and and shortened, shifted, or better explained paragraphs within the text.

**R2** *Page 1, Lines 13-14. How to distinguish climate drivers of variability for both modelled and measured isotopes?*
Thank you for pointing this out. Of course, we don't search for common drivers in both modelled and real world. We will rewrite the sentence as follows:
"... We systematically evaluate differences and commonalities between the standardized model simulation outputs. The goal is to distinguish climatic drivers of variability **for** modelled **isotopes and compare them to those of** measured isotopes. ..."
Action: Done.

**R2** *Page 2, Lines 20-21. Is it possible to show the formula of carbon isotope like oxygen isotope (line 19)?* We will add a definition in line 24, where we introduce the carbon isotopes. It will read as follows:

"...Oxygen and carbon isotopes ($\delta^{13}$C) are incorporated in calcite or aragonite matrices in accumulated growth layers and have long been used as proxies of terrestrial climate (Hendy, 1971). **For carbon isotopes, the** $\delta$ **notation is given as** $\delta^{13}$**C** $=$ $\left( \frac{\frac{^{13}\mathbf{C}}{^{12}\mathbf{C}} \mathbf{sample}}{\frac{^{13}\mathbf{C}}{^{12}\mathbf{C}} \mathbf{standard}} - 1 \right) \cdot 1000$ **‰ against V-PDB**....."

Action: We shifted the introduction of oxygen and carbon isotopic ratio definitions to the method section

*Page 2, Line 35. Please add the cave monitor work (Duan et al., 2016).*
Thank you for the suggestion. We will add the work to the section.

Action: Done.

**R2** *Page 2, Line 38. How to understand the "speleothem carbon isotopes can be easier to interpret than oxygen isotopes"? What's the easy explanation of the speleothem carbon isotopes?*

Thank you for pointing our this misleading sentence. We wanted to emphasis, that for specific caves, some proxies may be easier to interpret than others. Our statement is also meant the other way around, that oxygen may be easier to interpret than carbon isotopes in other caves. We will rewrite the statement as follows:

"... Depending on the specific site, **some proxies may be easier to interpret than others. As such,** speleothem carbon isotopes can **carry a more straightforward signal** than oxygen isotopes **where overlapping processes in specific regions can complicate interpretation** (Scholz et al., 2012; Ridley et al., 2015), especially during large climate changes such as the deglaciation (Genty et al., 2006). **Vise versa, carbon isotope sometimes need to be pre-constrained through the help of other proxies, e.g. $\delta^{18}$O to determine dominant processes (Fohlmeister et al., 2017).** Studies considering both isotopes profited from the isotopes' mutual information on fractionation processes and were able to disentangle the encoded climatic signal (Fohlmeister et al., 2017; Baker et al., 2017; Novello et al., 2019)...."

Action: Done.

**R2** *Page 3, Lines 10-15. What's the main conclusion from the previous model-data comparison? A detailed explanation is necessary to emphasize the motivation and innovation of this work.*

Previous model-data comparisons using the SISALv2 database do support the usage of the database to evaluate modelled $\delta^{18}$O in different time periods and to investigate different climatic features. Comas-Bru et al. (2019) found a consistency between observed and simulated changes in $\delta^{18}$O between ECHAM5-wiso and SISALv2. However, the simulation could underestimate some of these changes between the researched time periods (Mid-Holocene and Last Glacial Maximum). The study suggests that speleothems are under a large effect of site specific parameters which can contribute significantly to regional signals. Thus, they conclude that both mismatches between models and speleothems, and speleothem chronological and proxy uncertainties, are reasons to mainly focus on large-scale spatial patterns. In studies on isotopic fingerprints of major climate modes (such as monsoons, ENSO and PDO), Midhun et al. (2021) found that pseudo-stalagmites spatially correlated with signatures of ENSO and PDO using iCESM, and Parker et al. (2021) found that using ECHAM5-wiso and GISS-E1-R, relationships between speleothem $\delta^{18}$O and changes in circulation and precipitation were captured by speleothems in monsoon regions in Mid-Holocene, Last Interglacial and Last Glacial Maximum. Using iHadCM3, Bühler et al. (2021) found a fairly small time-mean spatial offset during last millennium, but lower speleothem $\delta^{18}$O variability than the simulated $\delta^{18}$O on interannual to decadal timescales. A lower temporal resolution of speleothem records and karst effects that smooth the $\delta^{18}$O signal suggests that data-model comparisons perform better on (multi-)decadal and longer timescales (Comas-Bru et al., 2019; Bühler et al., 2021; Midhun et al., 2021).

Following the suggestion raised, we have summarized the main conclusions from previous comparisons and connected the remaining knowledge gaps to our aim and motivation of our study more clearly. This paragraph now reads as follows:

"The Speleothem Isotope Synthesis and Analyses (SISAL) working group has collected a large number of speleothem records globally and compiled the database SISALv2. It has been employed for model-data comparisons of the last glacial maximum, the Mid-Holocene, the last millennium, and the historical period using different models (iCESM: Midhun et al. (2021), iHadCM3: Bühler et al. (2021), ECHAM5-wiso: Comas-Bru et al. (2019); Parker et al. (2021) and GISS-E1-R: Parker et al. (2021)).. **The previous model-data comparisons supports the use of the database to evaluate modelled $\delta^{18}$O across different time periods, although speleothems have a lower $\delta^{18}$O variability than simulated $\delta^{18}$O on interannual to decadal timescales globally. However, a benchmarking study on model performance in simulating d18O, including multi-model comparison and model-data comparison with SISALv2 has not yet been performed.**"

Action: Done.

*R2* *Page 3, Lines 34-37. What's the main conclusion from the multi-model comparison? A detailed explanation is also necessary to emphasize the motivation and innovation of this work.*

We agree with the reviewer, that we can more strongly draw attention to the innovative aspects of our work. Along with the previous section, which summarizes conclusions of these multi-model studies, we will change the section as follows:

"... The second evaluation in the SWING2-intercomparison of isotope-enabled AGCMs in 2012 showed that model differences most likely arise from differences in processes that control atmospheric humidity (Risi et al., 2012). Conroy et al. (2013) found that models which realistically capture precipitation patterns in the tropics are not necessarily successful in simulating the isotopic composition of precipitation compared to measured data and vice versa, cautioning on always using multiple models when comparing to paleoclimate proxy records. All models that are used in this study have been part of the SWING2 assessment for the historical period in their current, previous, or atmosphere-only version. **The historical period multi-model comparison is, however, too short to analyse and compare multi-decadal to centennial isotopic variability.** Therefore, this multi-model comparison complements previous work (Jungclaus et al., 2017; Midhun and Ramesh, 2016; Conroy et al., 2013), through its focus on **how different models represent** SWI **and its variability on different timescales** over the entire last millennium. We aim to identify common model biases (Kageyama et al., 2018) globally and in different regions, as well as distinguish specific climate drivers for modelled isotope variability on decadal and longer timescales. ... "

Also, we will change the outline in the introduction to:

"...Here we will present a multi-model comparison of five isotope-enabled last millennium simulations: ECHAM5/MPI-OM (Sjolte et al., 2018), GISS ModelE2-R (Lewis and Legrande, 2015; Colose et al., 2016a,b), the iGCM version of the Community Earth System Model (CESM) (Stevenson et al., 2019; Brady et al., 2019), the iGCM version 3 of the Hadley Model (HadCM) (Bühler et al., 2021), and the water isotope-incorporated Scripps Experimental Climate Prediction Center's GSM (Yoshimura et al., 2008), with climate characteristics and forcings as depicted in Fig. 1 and listed in Tab. 1. **This allows, for the first time, for the joint intercomparison of stable water isotopologue variability in climate models and proxy archives in a time period dominated by natural forcing.**."

Action: We completely revised the introduction section to be more concise and more motivating based on both reviewers suggestions.

Also, we will emphasize this more in the conclusion:

"...**This joint intercomparison of stable water isotopologue variability in both models and speleothem data is the first dataset in a time period of natural forcing and allows for more future analysis by the scientific community.** Our analysis encourages the use of multi-model means whenever possible as already suggested by other studies (Colose et al., 2016a). From the point of model evaluation, the incorporation of different archives with higher resolution (e.g. corals, trees, ice cores as in the iso2k database (Konecky et al., 2020)) and with the help of improved proxy system models may provide further insight into why offsets between models can be so large regionally. From a speleothem perspective, within-cave and between-cave variability comparisons using both ...."

Action: Done.

**R2** *It is recommended to illustrate the ability of each model to simulate oxygen isotope in the introduction or Data section, which would help the readers to explain the differences among the models.*

We follow the reviewer's suggestion, and will add figure A3 and Fig. A2, which was also suggested by reviewer 1, to the supplement file in the revised version of the manuscript. This figure will clearly show each model's individual representation at the speleothem location. We will also refer to these shortly in data section 2.1, the results section 4.1, and the discussion section 5.1 in the updated manuscript. For differences between the models, we will also add the vertical resolution of each model to Table 1 for reference.

Action: We added Fig. A2 and Fig. A2 to the supplement and added information

[Figure]

Figure A2: Mean simulated $\delta^{18}O_{iw}$ across all latitudes for all simulations.

to the data and results section.

**R2** *The past millennium includes different climatic backgrounds (Medieval Warm Period, Little Ice Age, and Modern Warm Period), and the spatial distributions and main driving factors of simulated water isotopes and measured speleothem isotopes may be different under warm and cold backgrounds. Comparison analyses in different climatic backgrounds are suggested.*

We agree with the reviewer, that both simulated and measured SWI will be different under different background states. While signatures of LIA-cooling or MCA-warming exist on a regional scale (McDermott et al., 2001), there is no global coherence of cold or warm periods over the Common Era (Neukom et al., 2019). Modelled global mean isotopic signatures of the models used in this analysis maximally differ by 0.1‰between the LIA and the MCA, and not even all models agree in the direction of the change. The intra-model comparison between the two periods are also still within the general inter-model range of global mean isotopic concentration which is well above 2‰. Other model-data comparisons also didn't include specific analysis on the LIA and MCA (Werner et al., 2016). Regional studies with spatially higher resolved models are necessary to analyse if signatures are visible. The current anthropogenic warming is of course visible in both model and data (Shukla et al., 2019), which is however not part of this study, where we only analyzed the last millennium until 1850CE. Different climatic backgrounds e.g. between LGM and the Holocene are also visible in both model and data and offsets and biases is analysed in multiple studies (Comas-Bru et al., 2019; Tierney et al., 2020;

[Figure]

Figure A3: Speleothem $\delta^{18}O_{dweq}$ and simulated $\delta^{18}O_{iw}$ in a) ECHAM5-wiso, b) GISS-E2-R, c) iCESM, d) iHadCM3, e) isoGSM, and f) multi-model mean.

Parker et al., 2021). We will add these thoughts to our discussion.

Action: We added the following paragraph to the discussion:

Action: "...The impact of different climatic backgrounds on the $\delta^{18}O$ signal in speleothem records or paleoclimate simulations in time periods such as the LGM of the Holocene have been studied extensively (e.g. Comas-Bru et al., 2019; Tierney et al., 2020; Parker et al., 2021). Periods of documented warmer and colder periods within the last millennium are for example the Little Ice Age (1550-1850 CE) or the Medieval Climate Anomaly (850-1250CE) (definitions from ?). We note that neither the global mean surface temperature, nor the simulated $\delta^{18}O$ or the global $\delta^{18}O$ as recorded by the speleothems, showed significant changes to the mean state on a global scale within the described periods within the last millennium (results not shown). This is in line with Neukom et al. (2019) who found no global coherence of cold or warm periods over the Common Era, even though local changes are observable (e.g. McDermott et al., 2001). ..."

**R1** Page 4 lines 17-26: The objectives of this paper are currently in the form of somewhat run-on sentences. Readers may understand them more clearly if they are organized more effectively. For example, one possible way to reorganize could be: "With this study, we aim to contribute to the understanding of both model and data: 1) How do different simulations model oxygen isotopes in the hydrological cycle and how do they compare to archived speleothem data? 2) What processes influence speleothem isotopic

*composition and what effects of variability can be captured and later analyzed?"*

Thank you for this clarification. We will add an abc-enumeration, to not confuse the reader with the following text that starts with "We first..." and "In a second step...", as follows:

"...With this study, we aim to contribute to the understanding of both model and data: **a) H**ow do different simulations model oxygen isotopes in the hydrological cycle and how do they compare to archived speleothem data? **b) What** processes influence speleothem isotope composition and what effects of variability can be captured and later analyzed? ..."

Action: Done.

***R1*** *Table 1: Definitions (can be brief) of GTOPO and ETOPO are missing from either the table caption or manuscript text.*

Relevant references and definitions to GTOPO and ETOPO will be added to table 1 in the revised manuscript (i.e. Gesch et al. (1999); NOAA National Geophysical Data Center (2009); Amante and Eakins (2009); National Geophysical Data Center (1993)).

Action: Done.

***R1*** *Page 4 Data section: There are many differences in the boundary conditions used between the five models and their setups. It would be helpful to add text on the impacts that these differences may have on the resulting simulations. This will be important in understanding how much (or how little) we can attribute the variations in each simulation to their underlying boundary conditions or if other factors play a more dominant role in their simulated differences.*

Thank you for this interesting thought. We will add a short paragraph on the impacts of the different forcings in the data section 2.1 as follows:
"... Their basic characteristics and boundary conditions are listed in Tab. 1. They are both used individually in the analysis, as well as by the ensemble mean of all models. Fig. 1 shows the climate as represented by the different models and external forcings used in the simulations. **Since SWING2, there has not been a consistent protocol for paleoclimate simulations with isotope enabled models. Hence, the simulations used in this study largely follow the PMIP3 Last millennium experiment protocol (Schmidt et al., 2011, 2012) with its proposed climate forcing reconstructions, with some variations in vegetation and orography. Of the external forcings used, differences in volcanic forcing may have the**

largest influence on differences between the simulations (Colose et al., 2016a; Schmidt et al., 2011), as different responses on larger eruptions may have a long term impact. Large eruptions can cause local anomalies to the mean state $\delta^{18}$O of up to $\pm 1.5$‰ (Colose et al., 2016a), hinting at the magitude of change that can be caused by different forcings. These volcanic eruptions are among the most prominent drivers of natural climate variability (Jungclaus et al., 2017). Compared to volcanic forcing, the choice in solar or orbital forcing has a less strong effect over time in the last millennium. Although the simulations do use different forcings based on different reconstructions which then act on different timescales, differences in response may not only arise from the forcings, but from the implementation in the models Jungclaus et al. (2017). ..."

We will also change "vegetation" in the table to "land cover" as it describes the forcing more precisely.

Action: Done.

**R1** *Figure 1: For Figure 1a, please state what the anomalies are relative to (i.e., what is signified by 0°C? It appears to be ~1900 CE).*

Thank you for pointing this out. The anomalies are relative to the period of the last millennium (850-1850CE). We will add this in the caption.

Action: Done.

**R1** *Figure 1: Please describe more clearly what the difference is between the noisy background lines and the less variable darker colored lines in Figure 1a.*

Thank you for spotting this. The noisy background are the down-sampled values at cave location while the bold line are the down-sampled values with a 100 yr Gaussian kernel bandpass and smoothing from the R-package nest (`https://github.com/krehfeld/nest` Rehfeld et al. (2011); Rehfeld and Kurths (2014)).

Action: Done.

**R1** *Page 9 line 20: Are speleothem record values of d18Oc from the Last Millennium being converted into d18Odweq? If so, please describe how the past temperatures are calculated or inferred.*

As explained in the text on page 9 line 32-33, we use the annual mean modelled surface temperatures as a surrogate for measured cave temperatures, as these are often

not available especially in paleoclimate.

Action: In the new manuscript this section is revised on page 10 line 29-30.

**R2** *Page 12. Please check the description for Figure 3. It is difficult to find ECHAM5-wiso with more strongly depleted mid-latitude oceans than in the other simulations and iCESM and iHadCM3 with stronger depletion towards the poles compared to the other simulations; Modifying48 ‰to -8.48 ‰.*

Thank you for finding the missing minus sign. The first reviewer also noticed it and we will correct it in the revised manuscript. Also, following the suggestions of the first reviewer, we will revise the section as follows:

"... The global mean $\delta^{18}O_{iw}$ values are fairly similar in area-weighted global mean of 8.48‰ (90% CI: $-8.61$, $-8.36$) and $-8.41$‰ ($-8.62$, $-8.2$) for isoGSM and GISS-E2-R, respectively. The ECHAM5-wiso run is less depleted with a global $\delta^{18}O_{sim}$ mean of $-7.27$‰ ($-7.46$, $-7.09$),  **and** with  visibl**y** less  depleted mid-latitude oceans than in the other simulations. iCESM and iHadCM3 show a stronger depletion of $-9.39$‰ ($-9.51$, $-9.28$) and $-9.15$‰ ($-9.29$, $-9.01$) respectively, with **iCESM showing** stronger depletion **in the mid-latitudes and iHadCM3** towards the **Antarctic** compared to the other simulations. **Although GISS-E2-R shows strong depletion especially in the arctic region, the less depleted mid-latitudes dominate the global mean.** ..."

**R1** *Page 12 lines 16-18: The text states that iCESM and iHadCM3 show stronger depletion towards the poles compared to other models. From my view of Figure 3, I do not see this stronger depletion because I see that GISS-E2-R shows stronger polar depletion than either iCESM or iHadCM3.*

Thank you for spotting this. We double checked with latitudinal averages as shown in Fig. A2. We will change the section as follows:
"... The global mean $\delta^{18}O_{iw}$ values are fairly similar in area-weighted global mean of 8.48‰ (90% CI: $-8.61$, $-8.36$) and $-8.41$‰ ($-8.62$, $-8.2$) for isoGSM and GISS-E2-R, respectively. The ECHAM5-wiso run is less depleted with a global $\delta^{18}O_{sim}$ mean of $-7.27$‰ ($-7.46$, $-7.09$), but with clearly visible more strongly depleted mid-latitude oceans than in the other simulations. iCESM and iHadCM3 show a stronger depletion of $-9.39$‰ ($-9.51$, $-9.28$) and $-9.15$‰ ($-9.29$, $-9.01$) respectively, with **iCESM showing** stronger depletion **in the mid-latitudes and iHadCM3** towards the **Antarctic** compared to the other simulations. **Although GISS-E2-R shows strong depletion especially in the arctic region, the less depleted mid-latitudes dominate the global mean.** ..."

Action: Fig. A2 is added to the supplement and we changed the section as described above.

**R1** *Page 12 line 16: When interpreting d18Oiw over the ocean, is ECHAM5-wiso being more depleted than other models in the mid-latitude oceans potentially due to how much evaporation takes place here since the P – E weighting will likely assign a heavy role to E in determining amount weighting? Inclusion of a figure for global evaporation in the supplement, like SFigs 3 & 4 for temperature and precipitation, may help in answering this question.*

Thank you for the suggestion. In fig A4), we find, that ECHAM5-wiso is not exceptionally different in its evaporation. However, after more evaluation, we do find that ECHAM5-wiso simulates less precipitation in the mid-latitudes than the other simulations. ECHAM5-wiso is least depleted in heavy oxygen isotopes in the mid-latitudes in Fig A2, but deviates not too much from the model ensemble range. We add an additional figure of precipitation minus evaporation (Fig A5) to the supplement of the manuscript, to see differences between the simulations, that are affecting our analysis.

Action: We added Fig. A5 to the supplement and describe it in the discussion of model $\delta^{18}$O signatures in Sec. 5.1.

[Figure]

Figure A4: Simulated evaporation climatology (a-e) of the respective simulation: a) ECHAM5-wiso, b) GISS-E2-R, c) iCESM, d) iHadCM3, e) isoGSM).

**R2** *Page 13. It is better to indicate the latitude and longitude of the cave locations mentioned in the text.*

[Figure]

Figure A5: Simulated precipitation minus evaporation climatology (a-e) of the respective simulation: a) ECHAM5-wiso, b) GISS-E2-R, c) iCESM, d) iHadCM3, e) isoGSM).

Thank you for your suggestion. This will surely enhance information to readers, who want to compare with other caves. We will add longitude, latitude and elevation information to the cave sites.

Action: Done.

**R1** *Page 13 lines 2-4: I disagree with the statement that iHadCM3 deviates in its simulation of northern Africa from the other models, but that all other models agree with each other. From my view, Figure 3 shows very different results in northern Africa across all models.*

Thank you. The statement is indeed wrong and we will change the text as follows: "...Restricting the view to low- to mid- latitudes, the largest model data difference is in the area of the Sahara desert, the Arabian peninsula, the Indian peninsula, **and Siberia, where low humidity, high precipitation amount or high continentality are the driving local forces of $\delta^{18}$O**.

Action: Done.

**R1** *Page 13 lines 23-24: The text states that ECHAM5-wiso is the only model with a positive offset mean, but based on Fig. 4b it appears that isoGSM also has a positive offset mean? Please address this.*

The dashed lines in Fig. 4b represent the medians (0.28‰), however the simulation mean of isoGSM is negative in relation to the speleothem dataset (-0.17‰). Both mean

[Figure]

Figure A6: a-b) as Fig. 5 in the manuscript. c) shows red colors, wherever absolute correlation estimates to temperature are larger than absolute correlation estimates to precipitation and vice versa in blue.

and median are presented for the simulations and in their differences to the speleothem dataset. We do this to both include the full data (through the mean) and to have less impact of extreme values and skewed distributions (through the median). To clarify this better, we will change the text from the third sentence of the paragraph as follows: "The general distribution and **differences** between each model and speleothem data are shown as kernel density estimates (Fig. 4). **The full datasets are acknowledged through the mean value, whereas median values exclude skewed distributions and extremes.**".

Action: Done.

**R2** *Page 14, Figure 5. it is not enough to obtain the driver relationship from the correlation in Figure 5. There is also a high correlation between precipitation and isotopes in the high latitudes of the northern hemisphere in Figure 5. The further feedback or circulation analysis is suggested. Moreover, it is worth noting that the sign of correlations between simulated $\delta^{18}O_{sim}$ and temperature is consistent with many correlations between measured $\delta^{18}O_{speleo}$ and modelled temperature, but this is not same for precipitation. A possible reason is also welcome.*

We agree with the reviewer, that Fig. 5 is not enough to draw the conclusions. We revised the figure to Fig. A6, where we added Fig. A6c) compared to the original figure. Red colors indicate higher absolute correlation estimates to temperature, blue colors indicate higher absolute correlation estimates to precipitation. The patterns that we described are much better visible here. Temperature is still the main driver of isotope variability in the higher latitudes while precipitation dominates in the lower latitudes. We add, however, that precipitation also dominates isotope variability in the Antarctic surrounded by a dominant temperature zone in the Southern Ocean.

We stated the exact numbers for sign agreement between correlation estimates for the simulation and the speleothem isotopes further down the text and also in the discussion. We will however add more explanation in the results section. We change the section as

follows:

"... The inter-model comparison shows more agreement in the correlation fields to temperature than to precipitation, when focusing only on cave locations: the sign of correlation between $\delta^{18}O_{sim}$ and simulated temperature agree for three and more simulations at 60% of locations, and for four and more simulations even at 26% of locations. For precipitation on the other hand, only 11 % of locations agree in sign for three and more simulations, while it is only 1.1 % with agreement in four or more simulations. **The more uniform temperature response to external forcing may increase the total number of significant correlation estimates and thus also the number of locations that agree in sign.** ..."
Action: Done.

***R1*** *Figure 5 caption: It is slightly unclear what you mean here by the correlation. Is this the correlation of time-mean values in speleothems vs. models? Is it the time-varying mean? Clarifying this in the text will be beneficial.*

We will change the caption to enhance readability as follows:
"Correlations between **SWI** and **modelled** temperature (a) and precipitation (b) **time series** in each gridbox. The background shows the average over all 5 simulation **correlation estimates** between annual $\delta^{18}O_{iw}$ and simulated annual temperature per gridbox (a), and for precipitation (b). Crosses indicate gridboxes, where correlation **estimates** for four or more models **have** the same sign as the **averaged estimate over all** simulations. Symbols indicate the mean correlation of the simulated temperature (precipitation) to the recorded $\delta^{18}O_{speleo}$ at record resolution. Crossed circles mark those, where more than four models agree in the mean sign of the correlation to $\delta^{18}O_{speleo}$. Black circles indicate the location of those speleothems in the last millennium subset that show no significant correlation to any model."

Action: Done.

***R2*** *Page 15, Figure 6. The caption of Figure 6 misses the description of (b) and (d). Significance levels should be added when discussing correlations.*
We will adjust the caption as follows

"Speleothem $\delta^{18}O_{dweq}$ (**first row**) and $\delta^{13}C_c$ (**second row**) against latitude (**first column**) and altitude (**second column**) as provided by the database. Linear regression lines are shown separately for northern and southern hemisphere in (a) and (c), while the $R^2$ and p corresponds to the global linear regressions. **Confidence bounds are 90 %.**"
Action: Done.

***R1*** *Page 15 line 15: The text states that there is a decreasing spread in d13C with*

*increasing altitude. Is this result robust? It looks to me like there is instead decreasing data density with increasing altitude, which would suggest that this result is not robust.*

We agree, and we also discuss this in the Discussion chapter, however you are right, that we should already point this our earlier. We will adjust the sentence as follows:
 "...However, the spread in $\delta^{13}C_c$ appears to decrease with increasing altitude (Fig. 6d), **although under decreasing data density**. ..."

Action: Done.

***R1*** *Page 16 line 5: The results indicating that d13C is more enriched with altitude are described as "results not shown". It would be great if these results were shown in the supplemental.*

A figure of scatter plots between the two isotopes and altitudes for the separate latitudinal bands will be added to the revised supplement file (Fig. A7). In the text "results not shown" will be replaced by "SFig. X".
 Action: Done.

[Figure]

Figure A7: Speleothem $\delta^{18}O_{dweq}$ and $\delta^{13}C_c$ against altitude as provided by the database.

***R1*** *Page 15 lines 4-5 and Pages 16 lines 7-8 & 17 lines 1-2: With these summary statements, please acknowledge existing literature to claim that, as expected or not as expected, you see these specific literature-established relationships (i.e., strong relationship with temperature) in your analysis.*

We set the results into perspective of existing literature in the discussion section - specifically the summary of pages 16 (lines 7-8) & 17 (lines 1-2) are then later discussed on page 21 (line 1-23). For the summary of page 15 (line 4-5), which is discussed on age 21 (24-33), we will add existing literature for perspective as follows in the discussion section:

"...For all simulations, temperature variability was the dominant driver in $\delta^{18}O_{sim}$ at high latitudes and precipitation variability at low latitudes (Fig. 5). **However, local and regional climate dynamics, such as landward moisture transport and ice sheet changes can mask and alter these relationships, as found for simulated isotopes in GISS-E2-R in a global study by LeGrande and Schmidt (2009).** At the cave sites, model-internal regional variability as well as the records' age uncertainties substantially decreased correlation estimates. ..."

We will additionally add short summaries in the results section:

For Page 15 line 4-5: "...The data suggests that two main drivers for 18O can be distinguished in specific regions - temperature is dominant in the high latitudes, while precipitation appears to be the main driver in the low latitudes, which is what we expected following the principles established by Dansgaard (1964)."

For Pages 16 lines 7-8 & 17 lines 1-2 : "... The spatial testing shows globally strong relationships between $\delta^{18}O_{dweq}$ to environmental factors, in particular to altitude, temperature, precipitation, and evaporation, **which is in line with previous studies (for example Comas-Bru et al., 2019; Baker et al., 2019)**. The spatial relationships between speleothem entity mean $\delta^{13}C_{speleo}$ and meteorological variables from model ensemble mean (Fig 8) only show clear relationships in the extratropical region, but not on a global scale. **This indicates more local influences as by Fohlmeister et al. (2020).**

Action: Done.

***R1*** *Page 20 lines 9-10: The tone of this sentence could be softened because as it stands the statement is probably too strong considering all of the other factors that could also be at play. I find that the word choice "likely" helps to soften the tone in statements like this.*

Thank you. We will add this as follows:

"...We found that the mean $\delta^{18}O_{sim}$ fields show global differences of 2.12 ‰ between the models, that could mostly **likely** be attributed to the global mean temperature differences 1.8 K between the models. ..."

Action: Done.

***R2*** *Page 20, lines 11-12. and Page 21 lines 24-25. It is too arbitrary to obtain the conclusion of the "major driver" for the climate model study.*

We agree with the reviewer, that some passages are not concluded detailed enough.

The passages will also be enhanced through the added evidence in the revised Fig. A6. We change the section as follows:

"... Similarly, most of the strong regional differences in $\delta^{18}O_{sim}$ between models could be explained by regional differences in simulated temperature (SFig. 3)..."

"... For all simulations, temperature variability was the dominant driver in $\delta^{18}O_{sim}$ at high latitudes and precipitation variability at low latitudes **and parts of the Antarctic**(Fig. 5c and **SF.5**). ..."

Action: Done.

**R2** *Page 21, line 27. What is "cave locations for 3 and more simulations"? Is it "3 or more simulated cave locations"?*
Sorry for the misleading formulation. We meant cave locations for $\geq 3$ simulations and will adjust the sentence accordingly.

Action: Done.

**R2** *Page 22, lines 34-35. A possible reason is welcome.* We will change the section as follows:
"... We found that 86% of speleothems have a significant temporal correlation between speleothem oxygen and carbon isotopes, with 47% even showing strong significant (anti-) correlations of $|c| > 0.5$. **High co-variability between both isotopes can either be caused by kinetic fractionation processes (Hendy, 1971) in the cave environment or may be externally forced. For example, (Fohlmeister et al., 2017) studied a stalagmite in a very arid region and found strong correlation between the isotopes. They**  attribute increased correlation to times of strong variations in cave-internal processes triggered by variations of external conditions. This simultaneity agrees with our findings that generally no extreme event in isotopes precedes the other, which can, however, also be attributed to low sampling resolution. **More local cave monitoring studies are necessary to potentially exclude kinetic fractionation effect as the dominant driver.** ..."

Action: Done.

**R1** *Page 23 lines 23-29: The present wording makes it seem like this paragraph contradicts itself, even though that is not the case. When stating "d18Osim showed that*

*cave locations are in general suitable to detect climatic changes due to volcanic or solar forcing", this could easily be erroneously interpreted as saying "caves are generally suitable…" I recommend changing the language to something like the following: "d18Osim showed that modeled isotopic values can generally detect climatic changes…"*

Thank you for this clarification. Your recommendation, however, does not emphasize enough, that we also mean the locations where the caves are set, which we think is important too. We will change the section as follows and hope to resolve possible erroneous interpretations with it:

"...Summarizing, the comparison to **modelled values** showed that cave locations in this study are in general suitable to detect $\delta^{18}O_{sim}$ **variations due to modelled** climatic changes **as reactions on** changes in volcanic of solar forcing. ..."

Action: Done.

**R2** *Page 24, lines 22-23. What is the evidence to support this conclusion?*

We thank the reviewer for raising this question. To clarify our statement further, we will refine the specific paragraph in the conclusion as follows:

"... We presented a multi-model comparison over five last millennium isotope-enabled simulations (ECHAM5-wiso, GISS-E2-R, iCESM, iHadCM3 and isoGSM) and compared their representation of isotopic signatures in mean and variability to paleoclimate data from a large speleothem database (a last millennium subset of SISALv2). We found that $\delta^{18}O_{sim}$ differed substantially between models on a regional scale as well as at speleothem cave sites, **which could in part be attributed to differences in simulated temperature, model biases in implementing water isotopes or topography, but also cave- and site-specific controls on speleothem isotopes. To compensate for these differences, we used multi-model means in spatial comparisons.** The isoGSM simulation showed the lowest absolute mean offset to the speleothems at cave locations, while all other simulations show only slightly higher offsets...."

Action: Done.

**R1** *Page 24 line 23: In the Conclusion, there is a statement that says, "This effect can be compensated by using the multi-model mean." In thinking about the recommendation for using a multi-model approach, I am left wondering if this recommendation is based on 1) that a multi-model mean is always a less extreme model value because it reduces local spatial biases from individual models, and thus generally provides a better matches to speleothem values as they are less extreme, or instead 2) that multi-model means mostly converge to the real speleothem value, regardless of whether it is an extreme value or not. It may be useful to address this nuance during discussion of the multi-model approach recommendation.*

Thank you for pointing out, that this can be understood in more than one way. We will clarify the section in the conclusion as follows:

"...We found that $\delta^{18}O_{sim}$ differed substantially between models on a regional scale as well as at speleothem cave sites. This could mostly be attributed to differences in modelled temperature between models. **Extreme model values that differ greatly from the rest can be compensated for** by using the multi-model mean **and thus reducing local spatial biases**. The isoGSM simulation showed the lowest absolute mean offset to the speleothems at cave locations, while all other simulations show only slightly higher offsets. ..."

Action: Done.

**Technical Comments/Corrections**

***R1*** *Page 10 line 19: "Annually weighted" is crossed out* Done.
Action: Done.

***R1*** *Page 12 line 14: Missing a minus sign?* Done.
Action: Done.

***R1*** *Page 13 line 5: The first supplemental figure mentioned in the text is SFig 3. Should SFigs 1 & 2 be mentioned prior to this?*

We will change the order of the supplementary figures according to their appearance in the text. Thank you for spotting this.
Action: Done.

***R1*** *Page 20 line 24: Missing a parenthesis?* Done.
Action: Done.

[revised manuscript text omitted]

---

## Author Response (AR2)

**Reply to the reviewers' comments: Investigating oxygen and carbon isotopic relationships in speleothem records over the last millennium using multiple isotope-enabled climate models (cp-2021-152)**

Janica C. Bühler, Josefine Axelsson, Franziska A. Lechleitner, Jens Fohlmeister, Allegra N. LeGrande, Madhavan Midhun, Jesper Sjolte, Martin Werner, Kei Yoshimura, and Kira Rehfeld

June 3, 2022

**Summary of changes**

We again thank the second reviewer for their comments and detailed reading. In response to the suggestions by the reviewer we

- added one sentence to the conclusion to clarify the interpretation of the conclusion,

- checked the reference format throughout the manuscript

A detailed response to the helpful remarks of the referee is given below.

**Reply to the second reviewer**

(Original report cited in italics)

*Dear authors of the manuscript "Investigating stable oxygen and carbon isotopic variability in speleothem records over the last millennium using multiple isotope-enabled climate models", I appreciate for a substantial effort in revising this manuscript. The current version is easier to understand than before, especially two very interesting questions in the discussion section. However, I still worry about the main conclusion that the temperature is dominant driver. Thus, I suggest that the manuscript should be accepted for publication after minor revision.*

Thank you for the second reviewing. We are pleased to read that our changes increased the readability of the manuscript.

**Major comments:**

*1) How to understand the significant temporal correlation between simulated temperature to $\delta^{18}O_{speleo}$? If I am right, the temperature effect on the speleothem $\delta^{18}O$ record is very small. Moreover, the equations 1 and 2 include the annual mean modelled surface temperatures in the drip water equivalent, which would increase the influence of temperature on the simulated oxygen isotope records.*

Thank you for the comment. For the correlation estimates between simulated variables and the isotope signal in speleothems in Figure 5, we make sure to always use the "raw" $\delta^{18}O$ measurements (denoted as $\delta^{18}O_{speleo}$) as given by the SISALv2 database and not the drip-water equivalents (denoted as $\delta^{18}O_{dweq}$) as described in equation (1) and (2). The higher number of speleothem entities that show significant correlation to temperature than to precipitation could be attributed to the more uniform response of modeled temperature to e.g. volcanic forcing between model ensemble runs compared to precipitation responses, which depend strongly on regional particularities. We have already highlighted this in the discussion section 5.2 (lines 5-16) on page 23.

*2) There are some divergences need to be checked. e.g. in the conclusion, "temperature was driving $\delta^{18}O_{iw}$ variability in high latitudes and precipitation in low latitudes. At cave site locations in particular, which are mostly located in low- to mid-latitudes, models agreed more on temperature being the driving factor of SWI variability than on precipitation." This implies that the models show that the temperature is more important than precipitation in all latitudes. Right?*

While large-scale hydroclimate patterns are well represented in general circulation models, they tend to struggle in realistically simulating regional hydroclimate particularities due to convection and cloud dynamic parameterizations. Hence, spatial and temporal consistency between models and proxy records has to be evaluated carefully (PAGESHydro2k-Consortium, 2017). A higher number of correlation estimates to temperature than to precipitation indicates a higher correlation to external forcing factors, as the temperature response to these forcings are more uniform (PAGESHydro2k-Consortium, 2017). Bühler et al. (2021) showed significant correlation between external volcanic forcing and simulated temperature but no significant correlation to precipitation for the iHadCM3 last millennium run, that is used in our multi-model ensemble. In our analysis on extreme synchronous events, volcanic forcing was detectable in $\delta^{18}O_{sim}$ on an annual basis but not on record resolution. To make it easier to interpret the results, we added the following explanation to our conclusion after the section that you cited (page 26, lines 23-24): "**However, temperature signatures in climate models are generally more uniform than those of precipitation, as these depend heavily on how models parameterize convection and cloud dynamics (PAGESHydro2k-Consortium, 2017).**".

**Specific Comment:**

*The format of the references should be checked. e.g. (Fohlmeister et al., 2017) studied..., and (PAGESHydro2k Consortium, 2017).*

Thank you for the careful reading. We scanned the document and changed citation styles where necessary.

**References**

Bühler, J. C., Roesch, C., Kirschner, M., Sime, L., Holloway, M. D., and Rehfeld, K.: Comparison of the oxygen isotope signatures in speleothem records and iHadCM3 model simulations for the last millennium, Climate of the Past, 17, 985–1004, 2021.

PAGESHydro2k-Consortium: Comparing proxy and model estimates of hydroclimate variability and change over the Common Era, Climate of the Past, 13, 1851–1900, https://doi.org/10.5194/cp-13-1851-2017, 2017.